# Benchmarking and evaluating the NASA Land Information System (version 7.5.2) coupled

with the refactored Noah-MP land surface model (version 5.0)

- 4 Cenlin He<sup>1</sup>, Tzu-Shun Lin<sup>1</sup>, David M. Mocko<sup>2,3</sup>, Ronnie Abolafia-Rosenzweig<sup>1</sup>, Jerry W. Wegiel<sup>2,3</sup>,
- 5 Sujay V. Kumar<sup>2</sup>

- 7 <sup>1</sup>NSF National Center for Atmospheric Research (NCAR), Boulder, Colorado, USA
- 8 <sup>2</sup>NASA Goddard Space Flight Center, Greenbelt, Maryland, USA
- 9 <sup>3</sup>Science Applications International Corporation, Greenbelt, Maryland, USA

12 Correspondence to: Cenlin He (cenlinhe@ucar.edu)

### Abstract

We integrate the refactored community Noah-MP version 5.0 model with the NASA Land Information System (LIS) version 7.5.2 to streamline the synchronization, development, and maintenance of Noah-MP within LIS and to enhance their interoperability and applicability. We evaluate and compare 5-year (2018-2022) global and regional benchmark simulations of LIS/Noah-MPv5.0 and LIS/Noah-MPv4.0.1 for a set of key land surface variables. Both models capture the spatial and seasonal distributions of observed soil moisture, latent heat (LH), snow water equivalent (SWE), snow depth, snow cover, and surface albedo, with similar bias patterns. Both models tend to underestimate soil moisture over wet soil regimes and overestimate over dry soil regimes, with slightly higher (≤ ~0.01 m³/m³ for global mean) soil moisture in LIS/Noah-MPv5.0 than LIS/Noah-MPv4.0.1 across most regions. The model bias patterns of LH overall follow those of soil moisture, while LIS/Noah-MPv5.0 has a lower LH across many non-polar regions than LIS/Noah-MPv4.0.1, which reduces the global mean LH bias from 0.99 W/m² to -0.39 W/m². The model SWE bias patterns are dominated by the precipitation and temperature forcing uncertainties, with slightly lower SWE values in LIS/Noah-MPv5.0 (global mean bias of -10.1 mm). The model bias patterns of

snow depth generally follow those of SWE. LIS/Noah-MPv4.0.1 consistently overestimates snow cover globally with a mean bias of 0.11, while LIS/Noah-MPv5.0 effectively reduces the overestimates across the global snowpacks with a mean bias of 0.07 because of updated snow cover parameters. Both models show widespread overestimates of surface albedo over mid-latitude and high-latitude regions but significant underestimates in the Sahara Desert and Antarctica. Overall, LIS/Noah-MPv5.0 outperforms or is similar to LIS/Noah-MPv4.0.1 in the evaluated land surface variables, except for slight degradation in simulated surface soil moisture and SWE. This study reveals possible model deficiencies, motivates future improvements in coupled canopy-snowpack-soil processes and input soil data, and points to the importance of considering observational and forcing data uncertainties in model evaluation.

#### 1. Introduction

Land processes play a profound role in the Earth and climate systems through altering surface water and energy balances and feedback to the atmosphere (Fisher and Koven, 2020; Blyth et al., 2021). Earth's land surface provides important boundary conditions for atmospheric processes and climate/weather predictions particularly at the subseasonal-to-seasonal (S2S) time scale (Koster and Walker, 2015; Benson and Dirmeyer, 2023). Furthermore, as climate changes, increasing climate/weather extremes (e.g., drought, flood, heatwave, and fire) and food-water security issues (e.g., agricultural production and irrigation management) are happening at the land surface, triggering key crises for the society (Sillmann et al., 2017; AghaKouchak et al., 2020). To tackle these critical land-related environmental issues, accurate land modeling systems are needed.

There have been substantial efforts in the past decades to develop and improve various land modeling systems (e.g., Dickinson et al., 1993; Liang et al., 1994; Chen et al., 1997; Ek et al., 2003; Oleson et al., 2010; Best et al., 2011; Niu et al., 2011; Haverd et al., 2018). Among them, the NASA Land Information System (LIS) is a widely used, established open-source framework for high performance land surface and terrestrial hydrology modeling as well as data assimilation (DA) of satellite and ground-based observations (Kumar et al., 2006; Peters-Lidard et al., 2007; Kumar et al., 2008a). The LIS system integrates different land surface models (LSMs), satellite and ground observations, and advanced computing and data management tools, to enable an interoperable environment that is applicable across different spatial and temporal scales. Various

model developments and applications using LIS have been conducted in the past decade, such as coupling with atmospheric models to improve weather predictions (Kumar et al., 2008b; Wu et al., 2016), DA of observed vegetation, snow, terrestrial water storage, albedo, and soil conditions to improve land surface modeling (Liu et al., 2015; Santanello et al., 2016; Kumar et al., 2016; Kumar et al., 2019; Kumar et al., 2020), and applications for hydrological predictions (Arsenault et al., 2020), food security (Hazra et al., 2023), and land analysis (Nie et al., 2024).
































LIS allows the use of an ensemble of LSMs, such as Noah (Chen et al., 1997; Ek et al., 2003), Noah-MP (Niu et al., 2011), CLM (Oleson et al., 2010), VIC (Liang et al., 1994), JULES (Best et al., 2011), and CABLE (Haverd et al., 2018). Among them, Noah-MP is one of the most commonly used state-of-the-art LSMs in the world (He et al., 2023a). Built upon the Noah LSM, Noah-MP has significant enhancements in representations of canopy-snow-soil-hydrology processes and interactions as well as capabilities of modeling human activity impacts (e.g., crop dynamics, irrigation dynamics, tile drainage, and urbanization). The multi-parameterization options of Noah-MP further allow for uncertainty analysis and model performance optimization/calibration based on multi-physics model ensembles (Li et al., 2020). Noah-MP has been serving as a key land component of various research and operational weather and hydroclimate models, such as the NOAA Unified Forecast System (UFS), the Weather Research and Forecasting (WRF) model, the U.S. National Water Model (NWM), the Model for Prediction Across Scales (MPAS), the Korean Integrated Model (KIM), and the Chinese Global-to-Regional Integrated Forecast System (GRIST). Because of its advantages, Noah-MP has been applied in numerous applications, including high-resolution climate modeling (Liu et al., 2017; Rasmussen et al., 2023), vegetation and soil DA (Kumar et al., 2019; Xu et al., 2021), climate extremes (Arsenault et al., 2020; Kumar et al., 2021; Abolafia-Rosenzweig et al., 2022a, 2023, 2024a), snowpack and hydrology (He et al., 2019; Jiang et al., 2020; Hazra et al., 2023), agriculture and groundwater (Barlage et al., 2021; Zhang et al., 2023, 2025), and urban climate (Xue et al., 2024, 2025).

Recently, the community Noah-MP has undergone a substantial code modernization effort (version 5.0) to improve its modularity and interoperability (He et al., 2023b), with many physics updates and bug fixes compared to the versions 3.6 and 4.0.1 in LIS. These two earlier Noah-MP versions in the current LIS (version 7.5.2) were implemented by manually replicating the Noah-MP source code and updating LIS/Noah-MP interface and drivers, which does not allow easy

model upgrades and hence leads to a long-delayed version update compared to the community Noah-MP. Thus, in this effort, we describe the streamlining of the development and maintenance of Noah-MP in LIS to enable the seamless integration between LIS and the community Noah-MP version to further enhance the interoperability and applicability of both models. Specifically, we couple the refactored community Noah-MPv5.0 with the LIS framework through the GitHub submodule mechanism accompanied by developing a new LIS/Noah-MP interface, which provides a direct, automatic link between the two models' source codes. This integration will allow easy code updates, synchronization, and maintenance for the coupled LIS/Noah-MP framework. The second goal of this study is to evaluate and compare global and regional benchmark simulations between LIS/Noah-MPv5.0 and LIS/Noah-MPv4.0.1 for key land surface conditions. Such systematic benchmarking is needed to examine the realism of LIS/Noah-MP model simulations, quantify the gaps between modeling and observations, and identify key processes for future model enhancements. This study is a step toward establishing a "scorecard" type of practice for LSMs.

## 2. Model descriptions and simulations

# 2.1 NASA LIS

The LIS system is a land surface hydrology digital twin environment, with the development led by the Hydrological Sciences Laboratory at NASA's Goddard Space Flight Center. Because of its extensible and flexible software infrastructure, LIS allows customized land DA systems and multiple LSMs to be integrated, assembled, and reconfigured easily using shared plugins and standard interfaces. Currently, LIS is the land component for several Earth system models, such as the NASA Unified WRF (NU-WRF) model, and the key component of several land DA system (LDAS) such as Global LDAS (GLDAS), North American LDAS (NLDAS), the Famine Early Warning Systems Network (FEWS NET) LDAS (FLDAS), and the operational land DA analysis environment at the U.S. Air Force Weather (Eylander et al., 2022).

Specifically, the LIS software suite consists of three main components: (1) Land Data Toolkit (LDT; Arsenault et al., 2018), which handles the data-related requirements of LIS including land surface parameter processing, geospatial transformations, consistency checks, data assimilation preprocessing, and forcing bias correction; (2) Land Information System (LIS), which is the modeling system that encapsulates land and hydrological models, DA algorithms,

optimization and uncertainty estimation algorithms, and high performance computing (HPC) support; and (3) Land Verification Toolkit (LVT; Kumar et al., 2012), which is a model verification and benchmarking environment that can be used for enabling rapid prototyping and evaluation of model simulations by comparing against a large suite of in-situ, remote sensing, and model and reanalysis data products. More details can be found at the LIS website: https://lis.gsfc.nasa.gov/ (last access: November 25, 2024). In this study, we use the LIS version 7.5.2 (latest version at the time of this work) coupled with Noah-MP in benchmark simulations and the LVT for model evaluation.
































# 2.2 Integration of refactored Noah-MPv5.0 with LIS

In this study, we couple the LIS system with the refactored community Noah-MPv5.0 model through the GitHub submodule mechanism to streamline the synchronization of Noah-MP between the community version and the LIS version, which will simplify future code updates and maintenance of Noah-MP within LIS. The GitHub submodule mechanism (https://gist.github.com/gitaarik/8735255) allows (1) separated source code maintenance and updates for Noah-MP (by the Noah-MP team) and LIS (by the NASA/LIS team), and (2) convenient updates of Noah-MP inside LIS by updating the submodule link to a newer Noah-MP GitHub tag/branch version. Compared to the Noah-MPv4.0.1 model in LIS, the community Noah-MPv5.0 model includes several important updates and new features: (1) improved modularization with modern Fortran code structures, (2) new hierarchical model data types and structures, (3) enhanced subroutine interface and calling workflow based on the modularization and new data types, (4) new self-explanatory model variable and module names, and (5) model bug fixes and new physics schemes. The key bug fixes include updates in vegetation properties (e.g., bug fixes in vegetation fraction scaling treatments) and processes (e.g., bug fixes in canopy wind absorption parameters) as well as snowpack processes. The new physics schemes include improved parameters related to various snowpack processes, a new wet-bulb temperature-based snow-rain partitioning scheme, a new snow meltwater retention process, a new dynamic irrigation scheme, updated crop growth parameters, a new tile drainage scheme, a new canopy heat storage treatment, additional runoff schemes, and new capabilities to control the soil process timestep. More details of Noah-MPv5.0 features can be found in He et al. (2023b). The detailed Noah-MP physics and formulations are described in He et al. (2023c). The major code changes from Noah-MPv4.0.1 to

Noah-MPv5.0 described the model release available are in notes at: https://github.com/NCAR/noahmp/blob/master/RELEASE NOTES.md (last access: November 25, 2024). The key components we modify to couple LIS and Noah-MPv5.0 are the LIS/Noah-MP land model driver interface to create new input/output variable mapping, and the LIS initialization and master driver parts to leverage new modularized Noah-MP code modules. By taking advantage of the plugin and standard interfaces in LIS, the Noah-MPv5.0 model is also connected to other components of LIS, such as data assimilation, river routing, etc.
























156

157

158

159

160

161

162

#### 2.3 LIS/Noah-MP benchmark simulations

We conduct and evaluate two sets of benchmark simulations with LIS coupled with Noah-MP, including one set of regional simulations over the contiguous U.S. (CONUS) and one set of global simulations. Each set of the simulations includes one LIS/Noah-MPv4.0.1 simulation and one LIS/Noah-MPv5.0 simulation to compare their performance and quantify differences between versions. The regional simulations are conducted for 10 years (2013-2022) with a 5-year spin-up, which are driven by the hourly 0.125° North American Land Data Assimilation System (NLDAS-2) atmospheric forcing data (i.e., precipitation, surface temperature, surface pressure, surface specific humidity, wind speed, downward surface shortwave and longwave radiation). More details of NLDAS-2 data are described in Xia et al. (2012). The global simulations are conducted for 5 years (2018-2022) with a 5-year spin-up, and are driven by the global hourly ~10-km U.S. Air Force (USAF) atmospheric forcing reanalysis data (Kemp et al., 2022). More details of the forcing data (formerly known as AGRMET, AGRiculture METeorology) are described in Eylander et al. (2022). For all the simulations, the static land type map is from the Moderate Resolution Imaging Spectroradiometer (MODIS) satellite data (Figure 1), while the MODIS monthly climatological (2000-2008) leaf area index (LAI) and stem area index (SAI) are used (Yang et al., 2011). The static soil type map is from the State Soil Geographic (STATSGO)/Food and Agriculture Organization (FAO) soil database (FAO, 1991). For both LIS/Noah-MPv4.0.1 and LIS/Noah-MPv5.0 simulations, we adopt the same default Noah-MP physics options (see Appendix Table A1), which have been commonly used in previous Noah-MP applications to produce skilled model performance (He et al., 2023b). Model evaluations for both the regional and global simulations are focused on the 5-year period of 2018-2022.

**Figure 1**. MODIS land cover maps used for LIS/Noah-MP (a) global and (b) CONUS benchmark simulations.

#### 3. Reference data for model evaluation

We use a suite of reference datasets to evaluate the LIS/Noah-MP simulations of key land surface variables over the globe and CONUS, including soil moisture, latent heat flux (LH), snow water equivalent (SWE), snow depth, snow cover fraction, and surface albedo. Specifically, for surface soil moisture, we use the global daily 36-km Soil Moisture Active Passive (SMAP) version 8 Level 3 satellite data (O'Neill et al., 2021; https://nsidc.org/data/spl3smp/versions/8, last access: November 25, 2024). We also use the surface and root-zone soil moisture from the International Soil Moisture Network (ISMN) ground station hourly measurements (Dorigo et al., 2021; https://ismn.earth/en/, last access: November 25, 2024). The data quality control is done via LVT. For LH, we use the global 0.25° daily Global Land Evaporation Amsterdam Model (GLEAMv3.8a) reanalysis data (Miralles et al., 2011; https://www.gleam.eu/, last access: November 25, 2024) and the global 0.05° hourly FLUXCOM-X-BASE observation-based data (Nelson et al., 2024; https://gitlab.gwdg.de/fluxcom/fluxcomxdata, last access: July 6, 2025). For SWE and snow depth,

we use the daily 1-km NOAA National Weather Service's National Operational Hydrologic Remote Sensing Center (NOHRSC) Snow Data Assimilation System (SNODAS) data (Barrett, 2003; https://nsidc.org/data/g02158/, last access: November 25, 2024) and the global 0.1° ERA-5 land (ERA5-Land) reanalysis data (Muñoz-Sabater et al., 2021; https://www.ecmwf.int/en/era5land, last access: November 25, 2024). For snow cover fraction, we use the global daily 500-m **MODIS** Terra Cover version (Hall Snow data and Riggs, 2016: https://nsidc.org/data/mod10a1/versions/6, last access: November 25, 2024). For surface albedo, we use the global daily 0.05° MODIS Terra/Aqua merged data (Schaaf and Wang, 2021; https://lpdaac.usgs.gov/products/mcd43c3v061/, last access: November 25, 2024). For model evaluation, we re-map the reference gridded datasets to the LIS/Noah-MP model grids or bilinearly interpolate model values to in-situ measurement locations via LVT, which will likely introduce uncertainties to model evaluations. We also note that those reference datasets have their own uncertainties, which may impact model evaluation results.

215216






























## 4. Results and discussions

#### 4.1 Soil moisture

Figure 2 shows the global 5-year (2018-2022) mean surface soil moisture comparison between SMAP retrievals and LIS/Noah-MP simulations driven by the USAF forcing. Both LIS/Noah-MPv4.0.1 and LIS/Noah-MPv5.0 simulations capture the spatial and seasonal distributions of surface soil moisture globally (Figures 2 and S1), with similar bias patterns. Both simulations tend to underestimate in wet soil regimes (e.g., northern and eastern Canada, Amazon forests, northern Europe, tropical Africa, and southeast Asia) and overestimate in dry soil regimes (e.g., western US, west and east coasts of South America, southern and northern Africa, midlatitudinal Eurasia, and Australia), partially caused by the USAF precipitation forcing bias (Figure S2), except for northern Canada and southern Brazil which requires further investigation. We note that SMAP data quality is less reliable over regions with thick vegetation (e.g., Southeast US, Amazon rainforest, Congo Basin). The evapotranspiration (ET) biases caused by model deficiencies in plant hydraulics and root water uptake processes may also contribute to the soil moisture bias, as revealed by previous Noah-MP studies (Niu et al., 2020; Li et al., 2021). These global model bias patterns are consistent across all seasons (Figure S1). Due to the offset of model overestimates and underestimates across different regions, the global annual mean model bias is

small (0.003 m³/m³ for LIS/Noah-MPv4.0.1 and 0.008 m³/m³ for LIS/Noah-MPv5.0). Overall, LIS/Noah-MPv5.0 shows consistently higher surface soil moisture than LIS/Noah-MPv4.0.1 but the difference is small (Figure 2f), which is expected since there is no direct soil physics update but changes in snowpack (e.g., snow cover parameter updates) and vegetation processes (e.g., vegetation fraction scaling treatments) from Noah-MPv4.0.1 to Noah-MPv5.0.

**Figure 2.** Surface soil moisture ( $m^3/m^3$ ) comparison between SMAP retrievals and LIS/Noah-MP simulations driven by USAF forcing globally averaged during 2018-2022: (a) SMAP data, (b) LIS/Noah-MPv4.0.1 simulation, (c) LIS/Noah-MPv5.0 simulation, (d) LIS/Noah-MPv4.0.1 biases (model minus SMAP), (e) LIS/Noah-MPv5.0 biases (model minus SMAP), and (f) differences between LIS/Noah-MPv5.0 and LIS/Noah-MPv4.0.1 simulations. Grids with statistically significant differences (p 

Figure 3. Surface soil moisture (m³/m³) comparison between ISMN station measurements and LIS/Noah-MP simulations driven by USAF forcing globally averaged during 2018-2022: (a) ISMN data, (b) LIS/Noah-MPv4.0.1 simulation, (c) LIS/Noah-MPv5.0 simulation, (d) LIS/Noah-MPv4.0.1 biases (model minus ISMN), (e) LIS/Noah-MPv5.0 biases (model minus ISMN), (f) differences between LIS/Noah-MPv5.0 and LIS/Noah-MPv4.0.1 simulations, (g) LIS/Noah-MPv4.0.1 anomaly correlation, (h) LIS/Noah-MPv5.0 anomaly correlation, and (i) differences between LIS/Noah-MPv5.0 and LIS/Noah-MPv4.0.1 anomaly correlation. The global mean value is also provided in the lower right of each panel.

**Figure 4**. Same as Figure 3, but for root-zone soil moisture (m<sup>3</sup>/m<sup>3</sup>) evaluation.




















Over the CONUS, both LIS/Noah-MPv4.0.1 and LIS/Noah-MPv5.0 simulations driven by the NLDAS-2 forcing capture the spatial distribution of SMAP surface soil moisture with similar spatial bias patterns (Figure 5), which show model underestimates over wet soil regimes (e.g., the northwest coast and southeast and northeast U.S.) and overestimates over dry soil regimes (e.g., western and central U.S.). This is consistent with the global evaluation albeit using a different forcing dataset. LIS/Noah-MPv5.0 also produces consistently but slightly (0.007 m<sup>3</sup>/m<sup>3</sup>) higher soil moisture than LIS/Noah-MPv4.0.1 using the NLDAS-2 forcing (Figure 5f), similar to the results using the USAF forcing, revealing a robust difference pattern between the two model versions. The comparison with ISMN surface soil moisture data over the CONUS shows similar model bias patterns with those evaluated against SMAP (Figure 6), except for the northwest coast and Florida, where ISMN indicates dry soil regimes that are opposite to SMAP. This points to the importance of considering observational data uncertainty in model evaluation. The CONUS mean biases across all ISMN sites are 0.041 m<sup>3</sup>/m<sup>3</sup> and 0.047 m<sup>3</sup>/m<sup>3</sup> for LIS/Noah-MPv4.0.1 and LIS/Noah-MPv5.0, respectively. The CONUS mean anomaly correlation is about 0.6 for both models (Figure 6g-h), with slightly lower values particularly over many western U.S. sites for LIS/Noah-MPv5.0 than LIS/Noah-MPv4.0.1 (Figure 6i). The model bias pattern of root-zone soil moisture is similar to that of surface soil moisture but with larger underestimates at some central U.S. sites (Figure 7).

**Figure 5**. Same as Figure 2, but for evaluation of LIS/Noah-MP simulations driven by the NLDAS-2 forcing over the CONUS averaged during 2018-2022.

**Figure 6**. Same as Figure 3, but for evaluation of LIS/Noah-MP simulated surface soil moisture (m<sup>3</sup>/m<sup>3</sup>) driven by the NLDAS-2 forcing over the CONUS averaged during 2018-2022.

**Figure 7**. Same as Figure 4, but for evaluation of LIS/Noah-MP simulated root-zone soil moisture (m<sup>3</sup>/m<sup>3</sup>) driven by the NLDAS-2 forcing over the CONUS averaged during 2018-2022.

## 4.2 Latent heat flux






















Figure 8 shows the global 5-year (2018-2022) mean latent heat (LH) flux comparison between the GLEAM data and LIS/Noah-MP simulations driven by the USAF forcing. Both LIS/Noah-MPv4.0.1 and LIS/Noah-MPv5.0 simulations capture the spatial and seasonal LH distributions with similar bias patterns (Figures 8 and S3). The model LH biases are generally consistent with the surface soil moisture bias patterns (Figure 2), with the underestimated (overestimated) LH over regions with the underestimated (overestimated) soil moisture, except for northern Eurasia and northwest North America (Alaska and west Canada). Although LIS/Noah-MPv5.0 has a slightly higher soil moisture than LIS/Noah-MPv4.0.1 (Figures 2-4), it shows a lower LH (by up to ~15 W/m<sup>2</sup>) over some tropical and mid-latitude regions with the largest difference in the tropics, which reduces the global mean LH bias from 0.99 W/m<sup>2</sup> (LIS/Noah-MPv4.0.1) to -0.39 W/m<sup>2</sup> (LIS/Noah-MPv5.0). This difference in the two Noah-MP versions is mainly due to the code updates related to vegetation properties (e.g., bug fixes in vegetation fraction scaling treatments) and processes (e.g., added canopy heat storage treatment) which alters ET and LH (see Section 5 for discussion). The minor LH difference (up to ~5 W/m<sup>2</sup>) between the two model versions over the Antarctica and Greenland is mainly caused by updates in the glacier scheme that uses snowpack physics consistent with other land snowpacks in LIS/Noah-MPv5.0.

We note that the LH (or ET) reference data product also has nontrivial uncertainties which may confound model evaluations here (see Section 5 for detail).

**Figure 8**. Latent heat flux (W/m²) comparison between the GLEAM data and LIS/Noah-MP simulations driven by USAF forcing globally averaged during 2018-2022: (a) GLEAM3.8a data, (b) LIS/Noah-MPv4.0.1 simulation, (c) LIS/Noah-MPv5.0 simulation, (d) LIS/Noah-MPv4.0.1 biases (model minus GLEAM), (e) LIS/Noah-MPv5.0 biases (model minus GLEAM), and (f) differences between LIS/Noah-MPv5.0 and LIS/Noah-MPv4.0.1 simulations. Grids with statistically significant differences (p 

**Figure 9**. Same as Figure 8 but for evaluation of LIS/Noah-MP simulations driven by the NLDAS-2 forcing over the CONUS averaged during 2018-2022.

# 4.3 Snow water equivalent (SWE)

Figure 10 shows the global 5-year (2018-2022) mean SWE comparison for seasonal snowpack between ERA5-Land data and LIS/Noah-MP simulations driven by the USAF forcing. Both LIS/Noah-MPv4.0.1 and LIS/Noah-MPv5.0 simulations capture the spatial and seasonal SWE distributions with similar bias patterns (Figures 10 and S4). Both simulations tend to have much lower SWE (by up to 50 mm) in the Himalayas and west Canada than ERA5-Land, with slightly less SWE in eastern Russia, partially driven by overestimated surface temperature (Section 4.7). Both simulations have higher SWE than ERA5-Land in most other mid-latitude and high-latitude snowpacks, mainly driven by overestimated precipitation (Figure S2) and underestimated surface temperature (Figure S8). The global annual mean SWE biases are -10.1 mm and -13.2 mm for LIS/Noah-MPv4.0.1 and LIS/Noah-MPv5.0, respectively. Overall, LIS/Noah-MPv5.0 shows lower SWE than LIS/Noah-MPv4.0.1, particularly in spring when differences reach up to 25 mm (Figures 10f and S4) due to the updated snow cover parameters (He et al., 2021) that reduces snow cover fraction (Section 4.5) and enhances snow ablation particularly in spring through the positive surface albedo feedback. We note that the ERA5-Land SWE data also has uncertainties, which tends to overestimate SWE over mountainous areas (Monteiro and Morin, 2023).

**Figure 10**. SWE (mm) comparison between ERA5-Land and LIS/Noah-MP simulations driven by USAF forcing globally averaged during 2018-2022: (a) ERA5-Land data, (b) LIS/Noah-MPv4.0.1 simulation, (c) LIS/Noah-MPv5.0 simulation, (d) LIS/Noah-MPv4.0.1 biases (model minus ERA5-Land), (e) LIS/Noah-MPv5.0 biases (model minus ERA5-Land), and (f) differences between LIS/Noah-MPv5.0 and LIS/Noah-MPv4.0.1 simulations. Grids with statistically significant differences (p 

**Figure 11**. Same as Figure 10 but for SWE (mm) comparison between SNODAS and LIS/Noah-MP simulations driven by the NLDAS-2 forcing over the CONUS averaged during 2018-2022.

# 4.4 Snow depth

Figure 12 shows the global 5-year (2018-2022) mean snow depth comparison for seasonal snowpack between ERA5-Land data and LIS/Noah-MP simulations driven by the USAF forcing. Both LIS/Noah-MPv4.0.1 and LIS/Noah-MPv5.0 simulations reproduce the spatial and seasonal snow depth distributions with similar bias patterns (Figures 12 and S5). The snow depth bias pattern generally follows the SWE bias pattern (Figure 10) with global annual mean biases of ~0.06 m for both simulations, except for the lower snow depth over some regions with higher SWE in northern Canada and northern Russia compared to ERA5-Land. The snow depth difference (global mean of 0.003 m) between LIS/Noah-MPv4.0.1 and LIS/Noah-MPv5.0 is small (Figure 12f).

**Figure 12**. Snow depth (m) comparison between ERA5-Land and LIS/Noah-MP simulations driven by USAF forcing globally averaged during 2018-2022: (a) ERA5-Land data, (b) LIS/Noah-MPv4.0.1 simulation, (c) LIS/Noah-MPv5.0 simulation, (d) LIS/Noah-MPv4.0.1 biases (model minus ERA5-Land), (e) LIS/Noah-MPv5.0 biases (model minus ERA5-Land), and (f) differences between LIS/Noah-MPv5.0 and LIS/Noah-MPv4.0.1 simulations. Grids with statistically significant differences (p 

**Figure 13**. Same as Figure 12, but for snow depth (m) comparison between SNODAS and LIS/Noah-MP simulations driven by the NLDAS-2 forcing over the CONUS averaged during 2018-2022.

# 4.5 Snow cover fraction
























Although LIS/Noah-MPv4.0.1 and LIS/Noah-MPv5.0 simulations capture the spatial and seasonal snow cover distributions, they systematically overestimate snow cover globally relative to MODIS observations (Figures 14 and S6). This high bias in snow cover is particularly outstanding considering the underestimated SWE and snow depth (Figures 10 and 12), which has been a long-standing problem in Noah-MP (He et al., 2019; Jiang et al., 2020; Zhou et al., 2023). Specifically, LIS/Noah-MPv4.0.1 tends to overestimate snow cover across the global snowpack by up to 0.3 with a global mean bias of 0.11, while LIS/Noah-MPv5.0 reduces the snow cover overestimate particularly in northern high-latitudes and the Tibetan Plateau, which effectively reduces the global mean bias to 0.07. This bias reduction is attributable to the updated snow cover parameters in LIS/Noah-MPv5.0 (He et al., 2021). However, LIS/Noah-MPv5.0 still systematically overestimates snow cover over most mid-latitude and high-latitude snowpacks, which suggests the need for improved snowpack physics in Noah-MP (see Section 6 for discussion). The spatial heterogeneity of the snow cover change magnitude caused by the snow cover parameter updates may be due to several reasons: (1) The snow cover parameter updates are more effective for regions with snow depth less than about 0.3 m, since this is the most sensitive snow depth regime for snow cover calculations based on the parameterization used in Noah-MP (He et al., 2019); (2) The snow cover parameter updates are vegetation type dependent, so the

effectiveness of this change also depends on vegetation types; (3) Due to the positive surface albedo feedback, the snow cover change is more effective over ablation regions and periods; (4) The snow cover impact is further complicated by the spatial heterogeneity of SWE biases (Abolafia-Rosenzweig et al., 2025).

**Figure 14.** Snow cover fraction comparison between MODIS and LIS/Noah-MP simulations driven by the USAF forcing globally averaged during 2018-2022: (a) MODIS data, (b) LIS/Noah-MPv4.0.1 simulation, (c) LIS/Noah-MPv5.0 simulation, (d) LIS/Noah-MPv4.0.1 biases (model minus MODIS), (e) LIS/Noah-MPv5.0 biases (model minus MODIS), and (f) differences between LIS/Noah-MPv5.0 and LIS/Noah-MPv4.0.1 simulations. Grids with statistically significant differences (p 

**Figure 15**. Same as Figure 14, but for evaluation of LIS/Noah-MP simulations driven by the NLDAS-2 forcing over the CONUS averaged during 2018-2022.

### 4.6 Surface albedo

Figure 16 shows the global 5-year (2018-2022) mean surface albedo comparison between MODIS and LIS/Noah-MP simulations driven by the USAF forcing. Both LIS/Noah-MPv4.0.1 and LIS/Noah-MPv5.0 simulations capture the spatial and seasonal surface albedo distributions with similar bias patterns (Figures 16 and S7). LIS/Noah-MPv4.0.1 shows consistently overestimated surface albedo over most global regions by up to 0.05 or more, except for significant underestimates in the Sahara Desert and Antarctica which dominate the global mean bias (-0.02). This bias pattern is consistent across different seasons (Figure S7). Compared to LIS/Noah-MPv4.0.1, LIS/Noah-MPv5.0 shows an overall reduction of surface albedo across mid-latitudes and high-latitudes due to lower snow cover (Section 4.5), which reduces the high bias of surface albedo particularly in the mid-latitudes (Figure 16). The remaining albedo overestimates in LIS/Noah-MPv5.0 in the mid-latitude and high-latitude snowpacks are partially caused by the overestimated snow cover (Figure 14e) and also likely by the soil and vegetation albedo uncertainties. The systematic surface albedo underestimates in the Sahara Desert, Antarctica, and Greenland further indicate model biases in the background albedo for desert soil and glacier ice/snow albedo.

**Figure 16**. Surface albedo comparison between MODIS and LIS/Noah-MP simulations driven by USAF forcing globally averaged during 2018-2022: (a) MODIS data, (b) LIS/Noah-MPv4.0.1 simulation, (c) LIS/Noah-MPv5.0 simulation, (d) LIS/Noah-MPv4.0.1 biases (model minus MODIS), (e) LIS/Noah-MPv5.0 biases (model minus MODIS), and (f) differences between LIS/Noah-MPv5.0 and LIS/Noah-MPv4.0.1 simulations. Grids with statistically significant differences (p 

**Figure 17**. Same as Figure 16, but for evaluation of LIS/Noah-MP simulations driven by the NLDAS-2 forcing over the CONUS averaged during 2018-2022.

## 5. Discussion on resulting differences in two LIS/Noah-MP model versions

To summarize the evaluation metrics for all the investigated variables from both model simulations in this study, we adopted the International Land Model Benchmarking (ILAMBv2.7.2; Collier et al., 2018) package and applied it to our model simulations and reference datasets. Overall, the result (Figure 18) shows that LIS/Noah-MPv5.0 outperforms or is similar to LIS/Noah-MPv4.0.1 globally in the key land surface variables evaluated in this study, except for slight degradation in simulated surface soil moisture and SWE. In addition, we summarized all the bias values for different seasons and regions for both model simulations in Tables 1 and 2. The slightly degraded surface soil moisture simulation in LIS/Noah-MPv5.0 mainly comes from the degraded performance over northern and southern mid-latitudes, while the slightly degraded SWE in LIS/Noah-MPv5.0 is mainly caused by the degraded performance in the northern high-latitudes (Table 1). The soil moisture and SWE differences between the two model simulations are primarily caused by the model updates in vegetation processes (added canopy heat storage and bug fix of vegetation fraction scaling) and improved snow cover parameters.

**Figure 18.** Scorecard-type comparison for LIS/Noah-MPv4.0.1 and LIS/Noah-MPv5.0 model performance in simulating key surface variables evaluated against the reference datasets used in this study based on the ILAMB tool.

**Table 1.** Model evaluation metrics for LIS/Noah-MPv4.0.1 and LIS/Noah-MPv5.0 simulations driven by the USAF forcing averaged during 2018-2022 on the global and regional scale. The values are the annual mean model bias (LIS/Noah-MP simulations minus reference datasets). The statistically significant difference between LIS/Noah-MP v4.0.1 and LIS/Noah-MPv5.0 simulations (p < 0.05 using a t-test for daily time series) are marked as bold font. The values in the parentheses are the annual mean absolute model biases. The seasonal biases are shown in Tables S1-S4.

|             | Glo    | bal  | low latitude<br>(30°S - 30°N) |      |        | rn mid-<br>udes<br>60°N) | northern high-<br>latitudes<br>(>60°N) |      | southern mid-<br>latitudes<br>(30 - 60°S) |      | southern high-<br>latitudes<br>(>60°S) |      |
|-------------|--------|------|-------------------------------|------|--------|--------------------------|----------------------------------------|------|-------------------------------------------|------|----------------------------------------|------|
| LIS/Noah-MP | v4.0.1 | v5.0 | v4.0.1                        | v5.0 | v4.0.1 | v5.0                     | v4.0.1                                 | v5.0 | v4.0.1                                    | v5.0 | v4.0.1                                 | v5.0 |

| Surface soil<br>moisture (m³/m³<br>compared to SMAP)       | 0.003<br>(0.076)    | 0.008<br>(0.078)    | -0.009<br>(0.065) | -0.002<br>(0.066)  | 0.020<br>(0.079)  | 0.025<br>(0.082)   | -0.013<br>(0.093)   | -0.009<br>(0.094)   | 0.028<br>(0.081)    | 0.036<br>(0.086)    | 1                 | 1                 |
|------------------------------------------------------------|---------------------|---------------------|-------------------|--------------------|-------------------|--------------------|---------------------|---------------------|---------------------|---------------------|-------------------|-------------------|
| Surface Soil<br>moisture (m³/m³<br>compared to ISMN)       | 0.062<br>(0.078)    | 0.067<br>(0.082)    | 0.027<br>(0.061)  | 0.036<br>(0.067)   | 0.062<br>(0.079)  | 0.068<br>(0.082)   | 0.119<br>(0.121)    | 0.121<br>(0.123)    | 0.049<br>(0.062)    | 0.051<br>(0.062)    | 1                 | 1                 |
| Latent heat flux (W/m² compared to GLEAM3.8a)              | 0.992<br>(6.802)    | -0.386<br>(7.273)   | 2.105<br>(10.740) | -2.759<br>(11.601) | 0.752<br>(7.994)  | -0.608<br>(8.127)  | -4.122<br>(5.541)   | -3.784<br>(5.731)   | 1.469<br>(9.369)    | -0.271<br>(9.627)   | 2.992<br>(3.105)  | 3.668<br>(3.692)  |
| Snow water<br>equivalent (mm<br>compared to ERA5-<br>Land) | -10.123<br>(22.444) | -13.237<br>(22.328) | -0.845<br>(0.951) | -0.878<br>(0.966)  | 0.715<br>(16.267) | -1.349<br>(15.898) | -45.177<br>(71.928) | -56.181<br>(72.276) | -10.804<br>(16.494) | -10.471<br>(16.311) | 1                 | 1                 |
| Snow depth (m compared to ERA5-Land)                       | -0.059<br>(0.076)   | -0.061<br>(0.079)   | -0.003<br>(0.003) | -0.003<br>(0.003)  | -0.019<br>(0.051) | -0.019<br>(0.052)  | -0.231<br>(0.255)   | -0.245<br>(0.268)   | -0.040<br>(0.050)   | -0.037<br>(0.050)   |                   |                   |
| Snow cover fraction<br>(compared to<br>MODIS)              | 0.112<br>(0.113)    | 0.069<br>(0.090)    | 0.001<br>(0.003)  | 0.000<br>(0.002)   | 0.149<br>(0.151)  | 0.118<br>(0.122)   | 0.234<br>(0.235)    | 0.108<br>(0.183)    | 0.020<br>(0.027)    | 0.015<br>(0.023)    | 1                 | 1                 |
| Surface albedo<br>(compared to<br>MODIS)                   | -0.018<br>(0.061)   | -0.033<br>(0.067)   | -0.016<br>(0.047) | -0.017<br>(0.046)  | 0.032<br>(0.052)  | 0.021<br>(0.045)   | 0.016<br>(0.052)    | -0.024<br>(0.072)   | 0.017<br>(0.034)    | 0.013<br>(0.032)    | -0.084<br>(0.089) | -0.100<br>(0.102) |

**Table 2.** Model evaluation metrics for LIS/Noah-MPv4.0.1 and LIS/Noah-MPv5.0 simulations driven by the NLDAS-2 forcing averaged over the CONUS during 2018-2022. The values are the mean model bias (LIS/Noah-MP simulations minus reference datasets). The statistically significant difference between LIS/Noah-MP v4.0.1 and LIS/Noah-MPv5.0 simulations (p < 0.05 using a t-test for daily time series) are marked as bold font. The values in the parentheses are the mean absolute model biases.

|                                                      | Annual            |                   | D                  | JF                | MA                  | AM                  | JJA               |                   | SON               |                   |
|------------------------------------------------------|-------------------|-------------------|--------------------|-------------------|---------------------|---------------------|-------------------|-------------------|-------------------|-------------------|
| LIS/Noah-MP                                          | v4.0.1            | v5.0              | v4.0.1             | v5.0              | v4.0.1              | v5.0                | v4.0.1            | v5.0              | v4.0.1            | v5.0              |
| Surface soil<br>moisture (m³/m³<br>compared to SMAP) | 0.000<br>(0.062)  | 0.008<br>(0.065)  | 0.025<br>(0.077)   | 0.035<br>(0.085)  | 0.003<br>(0.067)    | 0.008<br>(0.069)    | 0.006<br>(0.062)  | 0.013<br>(0.065)  | -0.010<br>(0.058) | -0.001<br>(0.062) |
| Surface Soil<br>moisture (m³/m³<br>compared to ISMN) | 0.041<br>(0.065)  | 0.047<br>(0.068)  | 0.041<br>(0.075)   | 0.051<br>(0.080)  | 0.024<br>(0.066)    | 0.029<br>(0.067)    | 0.043<br>(0.069)  | 0.049<br>(0.072)  | 0.047<br>(0.069)  | 0.054<br>(0.074)  |
| Latent heat flux (W/m² compared to GLEAM3.8a)        | -0.207<br>(9.135) | -2.302<br>(9.286) | -5.864<br>(7.014)  | -5.126<br>(6.385) | -0.575<br>(14.912)  | -3.498<br>(14.413)  | 9.476<br>(17.752) | 3.209<br>(14.815) | -4.017<br>(7.147) | -3.865<br>(7.904) |
| Snow water equivalent (mm                            | -4.173<br>(6.422) | -4.959<br>(6.369) | -5.083<br>(10.148) | -6.715<br>(9.961) | -10.246<br>(13.924) | -11.309<br>(14.061) | -0.700<br>(1.221) | -0.961<br>(1.051) | -0.643<br>(1.018) | -0.843<br>(0.930) |

| compared to<br>SNODAS)                        |                   |                   |                   |                   |                   |                   |                   |                   |                   |                   |
|-----------------------------------------------|-------------------|-------------------|-------------------|-------------------|-------------------|-------------------|-------------------|-------------------|-------------------|-------------------|
| Snow depth (m<br>compared to<br>SNODAS)       | -0.013<br>(0.020) | -0.015<br>(0.020) | -0.016<br>(0.036) | -0.020<br>(0.035) | -0.032<br>(0.040) | -0.033<br>(0.040) | -0.002<br>(0.003) | -0.002<br>(0.002) | -0.004<br>(0.005) | -0.004<br>(0.005) |
| Snow cover fraction<br>(compared to<br>MODIS) | 0.055<br>(0.058)  | 0.028<br>(0.037)  | 0.221<br>(0.227)  | 0.117<br>(0.137)  | 0.045<br>(0.049)  | 0.026<br>(0.046)  | -0.003<br>(0.003) | -0.003<br>(0.003) | 0.018<br>(0.026)  | 0.004<br>(0.018)  |
| Surface albedo<br>(compared to<br>MODIS)      | 0.031<br>(0.038)  | 0.023<br>(0.033)  | 0.072<br>(0.083)  | 0.030<br>(0.056)  | 0.022<br>(0.032)  | 0.016<br>(0.029)  | 0.024<br>(0.033)  | 0.023<br>(0.033)  | 0.031<br>(0.041)  | 0.026<br>(0.037)  |

The modeled LH and soil moisture assessments in Section 4 indicate a slightly higher soil moisture but lower LH over some mid-latitude (e.g., the eastern U.S.) and the tropics in LIS/Noah-MPv5.0 compared to LIS/Noah-MPv4.0.1. To further understand this seemingly conflicting model differences, we conducted a series of additional analyses.

First, to quantify the uncertainty of reference ET data products, we conducted additional model evaluations using the FLUXCOM-X-BASE (Nelson et al., 2024) data. The results indicate large inconsistency between the FLUXCOM-X-BASE and GLEAM data, where the model biases reverse the signs across many global regions particularly in the low-latitudes (Figure S9). For the CONUS, the bias sign also reverses in the northeastern U.S. and many parts of the western U.S. (Figure S10). Previous studies (Nelson et al., 2024) showed that FLUXCOM-X-BASE has consistently lower ET in evergreen tropics as well as the temperate and high latitudes of the Northern Hemisphere than GLEAM, whereas FLUXCOM-X-BASE has higher ET in the semiarid and arid ecosystems of the lower and middle latitudes. This is consistent with previous studies (e.g., Abolafia-Rosenzweig et al., 2021a) finding large disagreements across ET reference datasets in general. These results suggest that the modeled ET in this study falls into the range of observational uncertainty over many global regions and the uncertainty in ET reference data products can confound model assessments which should be accounted for in future studies.

Then, to assess the role of soil temperature change, we further analyzed the soil temperature differences between the two model simulations, which indicates a consistently higher soil temperature in LIS/Noah-MPv5.0 than LIS/Noah-MPv4.0 across all soil layers over the majority

of the globe except for polar regions (Figures S11-12), which hence is not a driver but rather a result of the decrease in LH.

Furthermore, our additional analyses indicate that the bug fix of vegetation fraction scaling in LIS/Noah-MPv5.0 dominates the impact on the ET (and LH) reduction, with minor opposite effects from the added canopy heat storage term which generally increases sensible and latent heat fluxes (Figure S19). Furthermore, the LH changes tend to be larger over regions with higher vegetation fraction (Figure S20), which underlines potentially large and heterogeneous impacts in response to this.

In addition, we quantified the differences in each of the modeled ET components between the two model versions and their biases by comparing with the GLEAM data. Using the CONUS region as an example, the results show that the lower LH in LIS/Noah-MPv5.0 over the eastern U.S. is mainly caused by the lower plant transpiration and soil evaporation compared to LIS/Noah-MPv4.0.1, which outweigh the higher canopy-intercepted water evaporation (Figures S13-S15). The slightly lower LH in LIS/Noah-MPv5.0 over the western U.S. is dominated by the lower plant transpiration and canopy-intercepted water evaporation, which outweigh the higher soil evaporation. Overall, the generally opposite patterns in the western and eastern U.S. in these model differences in each ET component likely reflect the spatially heterogeneous impacts across water limited vs. non-water limited regimes, which needs further investigation. These patterns are generally consistent throughout the seasons (Figures S16-18), with stronger signals for plant transpiration and soil evaporation in spring and summer due to warmer temperature and higher solar radiation. Thus, the slightly higher soil moisture appears to be a result of the lower total ET in LIS/Noah-MPv5.0 compared to LIS/Noah-MPv4.0.1. Besides, the slightly higher soil moisture in LIS/Noah-MPv5.0 is also partially contributed by the updated snow cover parameters in LIS/Noah-MPv5.0 that lead to enhanced snow melting and hence increased soil moisture in winter, spring, and early summer.

For snowpack and surface albedo, LIS/Noah-MPv5.0 generally shows a lower SWE than LIS/Noah-MPv4.0.1 particularly during ablation periods, mainly due to the updated snow cover parameters in LIS/Noah-MPv5.0 resulting in lower snow cover and hence reduced surface albedo and subsequently enhanced melting. This triggers positive surface albedo feedback.

## 6. Implications for future model improvements
































The evaluation of global and regional benchmark simulations (Section 4) reveals several important Noah-MP model uncertainties and deficiencies, which calls for future model improvements.

First, the model biases in soil moisture and LH (Sections 4.1 and 4.2) partially reflect the inadequate representation of plant hydraulics and root schemes and/or too shallow soil column configuration (e.g., in the Amazon), which have also been highlighted by a few previous studies (e.g., Niu et al., 2020; Li et al., 2021; Bieri et al., 2025). Recently, Li et al. (2021) developed a new whole-plant hydraulics scheme for Noah-MP with observation-constrained parameters (Sun et al., 2024), which largely improves simulations of ET and terrestrial water storage (TWS) compared to the default soil hydraulics scheme in Noah-MP. Other studies (e.g., Niu et al., 2020; Bieri et al., 2025) developed dynamic root uptake schemes in Noah-MP that improve modeled soil moisture, ET, and TWS. These model updates have not been included in the community Noah-MPv5.0, which needs to be done in the future. Another possible model deficiency that could result in the LH bias is the canopy turbulence scheme. Noah-MP uses the Monin-Obukhov (M-O) similarity theory to compute momentum and heat exchange coefficients above and through the canopy, which however does not account for the canopy-induced turbulence in the roughness sublayer (RSL) and hence fails above and within dense forests (Bonan et al., 2018). Abolafia-Rosenzweig et al. (2021b) implemented and evaluated a unified RSL turbulence scheme throughout the canopy in an earlier Noah-MP version, which demonstrates the potential of improving modeled surface heat fluxes. We are currently working on a comprehensive assessment of this RSL canopy turbulence scheme in Noah-MPv5.0 across global FLUXNET sites. However, we note that the satellite soil moisture data has large uncertainties over dense forests. In addition, the input soil texture data could also impact the modeled soil moisture and hence ET. Li et al. (2024) recently developed a global 1-km high-quality datasets for key land surface parameters (including soil texture), and we plan to test the effect of using this new dataset in Noah-MP simulations in our next step.

Second, the model biases in snowpack, including SWE, snow depth, and snow cover, reveal inadequate treatments of snow physics. For example, the SWE underestimates over midlatitude mountains (e.g., the Himalayas and western U.S. high mountains) could be caused by the snow ablation bias in the model, in addition to the precipitation and temperature forcing

uncertainty (Section 4.3). He et al. (2021) found that Noah-MP tends to melt snow faster than observations in some western US mountain areas, likely due to wind and solar radiation forcing biases and/or model deficiencies in above-snowpack turbulence, canopy radiative transfer, and snow albedo. Recently, Lin et al. (2025) coupled Noah-MPv5.0 with a widely used physical snow albedo scheme, SNICAR-ADv3 (Flanner et al., 2021; He et al., 2024a), and found improved snow albedo relative to the default semi-empirical snow albedo scheme in Noah-MP. This snow albedo physics update will be included in the next Noah-MP major version release. The snow depth bias is not only driven by the SWE bias but also by uncertainty in snow compaction processes. A recent study (Abolafia-Rosenzweig et al., 2024b) enhanced the Noah-MP snow compaction parameterization constrained by in-situ measurements across ~800 SNOTEL sites in the western U.S., which is being currently transferred to the Noah-MPv5.0 (https://github.com/NCAR/noahmp/pull/148; last access: November 24, 2024). In addition, a new flexible framework was recently developed to couple the LSMs (including Noah-MPv4.0.1) in LIS with a physical snow model, Crocus, which shows promising improvements in modeling snow depth and SWE (Navari et al., 2024). The systematically overestimated snow cover fraction in Noah-MP is a long-standing model problem, which has been investigated by several studies over different mountain regions (He et al., 2019; Jiang et al., 2020; Zhou et al., 2023). A number of improvements in the model snow cover parameterization have been proposed for the Tibetan Plateau (Jiang et al., 2020; Zhou et al., 2023) and the western U.S. (Abolafia-Rosenzweig et al., 2025). These solutions, however, need to be tested for global applications.































Third, the model biases in surface albedo, particularly over the Sahara Desert and glaciers, suggest possible deficiencies in background desert soil albedo and glacier albedo. Currently, Noah-MPv5.0 assumes a uniform medium soil color everywhere, whereas using a spatially-varying soil color map (Lawrence and Chase, 2007) tends to reduce Noah-MP surface albedo particularly over the desert (Michael Barlage, personal communication), which will be tested in NoahMPv5.0 together with the aforementioned Li et al. (2024) global 1-km input datasets. To improve glacier modeling, Eidhammer et al. (2021) coupled the Crocus snow/ice model with Noah-MP within the WRF-Hydro framework, which reproduces the observed glacier surface albedo and mass balance in Norwegian glaciers. Future Noah-MP model improvements need to also focus on glacier regions, which were less studied in previous Noah-MP applications. In addition, vegetation albedo (and

canopy radiative transfer) may also contribute to the surface albedo biases in Noah-MP, which however lacks systematic assessments in the literature and hence needs more future investigations.

#### 7. Conclusions

In this study, we integrated the refactored community Noah-MPv5.0 model with the NASA LIS system (version 7.5.2) through the GitHub submodule mechanism to streamline the synchronization, development, and maintenance of Noah-MP within LIS and to enhance the interoperability and applicability of both models. The GitHub submodule mechanism also allows for more rapid implementation of bug fixes as well as new versions of Noah-MP (such as including the new physics detailed in Section 5) into the LIS software framework. We systematically evaluated multi-year (2018-2022) global and regional (CONUS) LIS/Noah-MP benchmark simulations driven by the USAF and NLDAS-2 atmospheric forcing, respectively, for a set of key land surface variables. Overall, LIS/Noah-MPv5.0 outperforms or is similar to LIS/Noah-MPv4.0.1 globally in simulating the key land surface variables evaluated in this study, except for slight degradation in simulated surface soil moisture and SWE.

Specifically, both LIS/Noah-MPv4.0.1 and LIS/Noah-MPv5.0 simulations capture the spatial and seasonal distributions of observed surface and root-zone soil moisture, LH, SWE, snow depth, snow cover, and surface albedo, with similar bias patterns. For surface and root-zone soil moisture, model simulations tend to underestimate over wet soil regimes and overestimate over dry soil regimes, with slightly higher soil moisture in LIS/Noah-MPv5.0 than LIS/Noah-MPv4.0.1 across most regions. Due to the offset of model overestimates and underestimates across different regions, the global mean soil moisture biases of both models are relatively small.

For LH, the model bias patterns generally follow those of soil moisture, with the underestimated (overestimated) LH over areas with the underestimated (overestimated) soil moisture across most global regions. Although LIS/Noah-MPv5.0 has a slightly higher soil moisture than LIS/Noah-MPv4.0.1, it shows a lower LH across most non-polar regions, which reduces the global mean LH bias from 0.99 W/m² (LIS/Noah-MPv4.0.1) to -0.39 W/m² (LIS/Noah-MPv5.0).

For snowpack conditions, the model SWE bias patterns are dominated by the precipitation and temperature forcing uncertainties, with large SWE underestimates in the Himalayas, west Canada, and western U.S. mountains and overestimates in most other mid-latitude and high-

latitude snowpacks. The SWE biases are similar for both models, with slightly larger underestimates in LIS/Noah-MPv5.0 (global mean bias of -13.2 mm) than LIS/Noah-MPv4.0.1 (global mean bias of -10.1 mm). The model bias patterns of snow depth generally follow those of SWE, with a global mean bias of ~0.06 m for both simulations. For snow cover, LIS/Noah-MPv4.0.1 has a systematic large overestimate across the globe, even over regions with underestimated SWE, which is a long-standing Noah-MP problem. LIS/Noah-MPv5.0 with updated snow cover parameters effectively reduces the snow cover overestimates globally, decreasing the global mean bias from 0.11 to 0.07.

For surface albedo, both models show widespread overestimates over most mid-latitude and high-latitude regions partially due to the snow cover overestimate, and significant underestimates in the Sahara Desert, Greenland, and Antarctica, which dominate the global mean bias. Because of the reduced snow cover, LIS/Noah-MPv5.0 shows consistently lower surface albedo than LIS/Noah-MPv4.0.1, which degrades the global mean bias from -0.018 to -0.033.

The model evaluation in this study reveals several important Noah-MP uncertainties and deficiencies and motivates future improvements in model processes/components including plant hydraulics and dynamic root uptake, canopy turbulence and interaction with snowpack, input soil texture and color data, snow cover and albedo, glacier ice, and vegetation albedo (canopy radiative transfer).

#### Appendix A.

**Table A1**. Default Noah-MP physics options used in this study

| Noah-MP Physics                                     | Option | Description                                                   |  |  |  |  |
|-----------------------------------------------------|--------|---------------------------------------------------------------|--|--|--|--|
| dynamic vegetation option                           | 4      | use table LAI and maximum vegetation fraction                 |  |  |  |  |
| rain-snow partition option                          | 1      | Jordan (1991) scheme                                          |  |  |  |  |
| soil moisture factor for stomatal resistance option | 1      | Noah (soil moisture) (Ek et al., 2003)                        |  |  |  |  |
| ground resistance option                            | 1      | Sakaguchi and Zeng (2009) scheme                              |  |  |  |  |
| surface drag coefficient option                     | 1      | Monin-Obukhov (M-O) Similarity Theory (Brutsaert, 1982)       |  |  |  |  |
| canopy stomatal resistance option                   | 1      | Ball-Berry scheme (Bonan, 1996)                               |  |  |  |  |
| snow surface albedo option                          | 1      | BATS snow albedo (Dickinson et al., 1993)                     |  |  |  |  |
| canopy radiation transfer option                    | 3      | two-stream applied to vegetated fraction (Niu and Yang, 2004) |  |  |  |  |

| snow/soil temperature time scheme option  | 1 | semi-implicit; flux top boundary condition (Niu et al., 2011)          |
|-------------------------------------------|---|------------------------------------------------------------------------|
| snow thermal conductivity option          | 1 | Stieglitz scheme (Yen,1965)                                            |
| lower boundary of soil temperature option | 2 | Deep soil boundary temperature read from input file (Niu et al., 2011) |
| soil supercooled liquid water option      | 1 | No iteration (Niu and Yang, 2006)                                      |
| runoff option                             | 3 | Schaake scheme (Schaake et al., 1996)                                  |
| frozen soil permeability option           | 1 | linear effects, more permeable (Niu and Yang, 2006)                    |
| soil configuration option                 | 1 | use input dominant soil texture                                        |
| glacier treatment option                  | 1 | include phase change of glacier ice                                    |
| tile drainage option                      | 0 | No tile drainage                                                       |
| irrigation option                         | 0 | No irrigation                                                          |
| dynamic crop model option                 | 0 | No dynamic crop model                                                  |

717718

# Code and data availability

- 1. The data and scripts produced in this study is available at 720 https://doi.org/10.5281/zenodo.14567219 (He et al., 2025).
- 2. The LIS/Noah-MPv5.0 model code produced and used in this study is available at https://doi.org/10.5281/zenodo.14567646 (He et al., 2024b).

## **Author contribution**

- CH, JW, and SK proposed research idea. CH performed technical coding and model coupling with
- help of DM. TL performed model simulations and evaluations with contribution from CH, DM,
- and RA. CH drafted the paper with contribution from TL, DM, RA, JW, and SK.

731

## **Competing interests**

The authors declare that they have no conflict of interest.

733734

# Acknowledgements

- This study was supported by the NASA Grant #80NSSC24K0121. The NSF National Center for
- Atmospheric Research (NCAR) is a major facility sponsored by the National Science Foundation
- (NSF) under Cooperative Agreement #1852977. We would like to acknowledge computing
- support from the Casper system (https://ncar.pub/casper) provided by the NSF National Center for
- Atmospheric Research (NCAR), sponsored by the National Science Foundation. Any opinions,
- findings, conclusions, or recommendations expressed in this publication are those of the authors
- and do not necessarily reflect the views of the National Science Foundation or NASA.

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
