# Peer review of "Benchmarking and evaluating the NASA Land Information System (version 7.5.2) coupled"

_EGUsphere, 2024_

## Author Comment (AC1)

**Reviewer #1 (Anonymous):**

In this manuscript, the authors present the integration of a new version of the Noah-MP land surface model (Noah-MPv5.0) into the Land Information System (LIS). They investigate the performance of the new LIS/Noah-MPv5.0 system against the previous version of the system (LIS/Noah-MPv4.0.1), focusing on a number of hydrological states and fluxes. They find slightly degraded performance of the new system for most of the variables investigated. I think that this manuscript will eventually provide a useful contribution to the community and constitute a valuable resource for user of LIS. However, I would suggest a few changes before the manuscript is accepted for publication.

Response: We thank the reviewer for the positive feedback and the constructive comments, which substantially improve our manuscript quality. We have addressed all the comments point by point below. In particular, after we added the results and scorecard based on the ILAMB benchmark and evaluation tool as suggested, we found that the new LIS/Noah-MPv5.0 model, compared with the LIS/Noah-MPv4.0.1 model, improves global model performance in all the key surface quantities evaluated in this study, except for slightly degraded performance in surface soil moisture and snow water equivalent. Please see also below for the details.

Major comments:

1. In its current form, the manuscript demonstrates the differences between LIS/Noah-MPv5.0 and LIS/Noah-MPv4.0.1 in detail, but in my opinion falls short when it comes to the interpretation of these differences. From the perspective of LIS users, I think that an understanding of the model changes that lead to the observed differences would be valuable, especially in cases where the model performance is degraded. The 'Discussion' section could be a good place to include more of an interpretation, but as it stands, that section focuses mostly on future model development without having established that the suggested future changes would address the cause of the differences observed currently.

Response: Thank you for the constructive comment. As suggested, we have included more analyses and discussions in the Discussion section for the interpretation of the differences between LIS/Noah-MPv5.0 and LIS/Noah-MPv4.0.1, particularly for cases with degraded model performance and the issue of seemingly conflicting soil moisture and LH changes. We have further separated the original Discussion section into two new sections, with the new Section 5 discussing the model differences and causes and the new Section 6 discussing the future model development.

      To avoid duplicated responses here, for model differences related to soil moisture and LH issues, please see our detailed responses to each of your comments below in this document and the new Section 5 in the revised manuscript, which include the new analyses and results.

      For model differences related to snowpack and surface albedo, we have included additional discussions in Section 5 as follows: "*LIS/Noah-MPv5.0 generally shows a lower SWE than LIS/Noah-MPv4.0.1 particularly during ablation periods, mainly due to the updated snow cover parameters in LIS/Noah-MPv5.0 resulting in lower snow cover and hence reduced surface albedo and subsequently enhanced melting. This triggers positive surface albedo feedback.*"

2. For example, the authors note the somewhat contradictory increase in surface soil moisture and simultaneous decrease in latent heat flux. I would have like to see a more in-depth discussion for the cause of this behavior. Are there changes to the model processes or parameters that would explain this? Given that the radiative forcing (and wind forcing) is the same in both experiments, would it make sense to check whether there has been a change in the soil temperature? Or look at a map of LAI to investigate whether this is caused by the bug fix to the vegetation fraction scaling that is mentioned.

Response: Thank you for the comment. We have included more analyses and discussions on the possible causes (e.g., changes in model processes or parameters) for the cases with increases of surface soil moisture and simultaneous decreases in latent heat flux. As suggested, we have analyzed the changes in soil temperature, which generally increases in LIS/Noah-MPv5.0 compared to LIS/Noah-MPv4.0 (see figures below) and hence is not the driver for the LH decrease but rather a result of the decreased LH. We have also analyzed the impact of the vegetation fraction scaling bug fix, which leads to changes in simulated ET and hence is an important contributor to the changes in soil moisture and LH in LIS/Noah-MPv5.0 compared to LIS/Noah-MPv4.0. We have also analyzed each of the modeled ET components to further understand the changes in total ET (and LH). We have included a detailed discussion in Section 5 ("Discussion" section) as follows:

"*The modeled LH and soil moisture assessments in Section 4 indicate a slightly higher soil moisture but lower LH over some mid-latitude (e.g., the eastern U.S.) and the tropics in LIS/Noah-MPv5.0 compared to LIS/Noah-MPv4.0.1. To further understand this seemingly conflicting model changes, we conduct a series of additional analyses.*"

(1) To assess uncertainties in LH reference data product, we have included the comparison with another soil-moisture independent ET product (FLUXCOM-X-BASE; Nelson et al., 2024) developed based on global FLUXNET flux measurements. We found that the data difference between GLEAM and FLUXCOM is large enough to reverse the model LH bias signs over many global regions, which can confound model assessments in this study. To avoid duplicated responses here, please refer to our response to your Major Comment #3 below for details.

(2) To assess the role of soil temperature change, we have analyzed the soil temperature difference between LIS/Noah-MPv5.0 and LIS/Noah-MPv4.0, and included the following discussions and figures in Section 5:

"*To assess the role of soil temperature change, we further analyzed the soil temperature differences between the two model simulations, which indicates a consistently higher soil temperature in LIS/Noah-MPv5.0 than LIS/Noah-MPv4.0 across all soil layers over the majority of the globe except for polar regions (Figures S11-12), which hence is not a driver but rather a result of the decrease in LH.*"

[Figure]

**Figure S11**. Simulated multi-year (2018-2022) annual mean soil temperature from LIS/Noah-MPv4.0.1 (left column), LIS/Noah-MPv5.0 (middle column), and their differences (right column) across four soil layers from the top (layer 1; first row) to the bottom (layer 4; fourth row). The LIS/Noah-MP simulations are driven by the USAF forcing globally. For glacier regions, the temperature is for glacier ice.

[Figure]

**Figure S12**. Same as Figure S11 but for LIS/Noah-MP simulations driven by the NLDAS-2 forcing over the CONUS.

(3) To further understand whether the LH change is due to other model updates in LIS/Noah-MPv5.0, including the added canopy heat storage term and bug fix to vegetation fraction scaling, we have conducted additional analyses and added discussions in Section 5 as follows:

"*Additional analyses indicate that the bug fix of vegetation fraction scaling in LIS/Noah-MPv5.0 dominates the impact on the ET (and LH) reduction, with minor opposite effects from the added canopy heat storage term which generally increases sensible and latent heat fluxes (Figure S19). Furthermore, the LH changes tend to be larger over regions with higher vegetation fraction (Figure S20), which underlines potentially large and heterogeneous impacts in response to this.*"

[Figure]

**Figure S19**. Latent heat (LH) and sensible heat (SH) flux changes due to the added canopy heat storage (first row) and all model updates (second row) in 2018 August.

[Figure]

**Figure S20**. Global and CONUS vegetation fraction (a-b) used in the model simulations, and the latent heat (LH) differences (c-d) between LIS/Noah-MPv5.0 and LIS/Noah-MPv4.0.1.

(4) We further analyzed the difference in each of ET components between LIS/Noah-MPv4.0.1 and LIS/Noah-MPv5.0 to better understand how LH responds to model changes between the two versions. We have included additional discussions and figures in Section 5 as follows:

*"We quantify the differences in each of the modeled ET components between the two model versions and their biases by comparing with the GLEAM data. Using the CONUS region as an example, we find that the lower LH in LIS/Noah-MPv5.0 over the eastern U.S. is mainly caused by the lower plant transpiration and soil evaporation compared to LIS/Noah-MPv4.0.1, which outweigh the higher canopy-intercepted water evaporation (Figures S13-S15). The slightly lower LH in LIS/Noah-MPv5.0 over the western U.S. is dominated by the lower plant transpiration and canopy-intercepted water evaporation, which outweigh the higher soil evaporation. Overall, the generally opposite patterns in the western and eastern U.S. in these model differences in each ET component likely reflect the spatially heterogeneous impacts across water limited vs. non-water limited regimes, which needs further investigation. These patterns are generally consistent throughout the seasons (Figures S16-18), with stronger signals for plant transpiration and soil evaporation in spring and summer due to warmer temperature and higher solar radiation. Thus, the slightly higher soil moisture appears to be a result of the lower total ET in LIS/Noah-MPv5.0 compared to LIS/Noah-MPv4.0.1."*

[Figure]

**Figure S13**. Comparison of latent heat flux (W/m²) due to soil evaporation between the GLEAM data and LIS/Noah-MP simulations driven by the NLDAS-2 forcing over the CONUS averaged during 2018-2021: (a) GLEAM3.8a data, (b) LIS/Noah-MPv4.0.1 simulation, (c) LIS/Noah-MPv5.0 simulation, (d) LIS/Noah-MPv4.0.1 biases (model minus GLEAM), (e) LIS/Noah-MPv5.0 biases (model minus GLEAM), and (f) differences between LIS/Noah-MPv5.0 and LIS/Noah-MPv4.0.1 simulations. Grids with statistically significant differences (*p* < 0.05) are shown with gray dots in panels (d)-(f). The statistical significance over each grid is computed using daily time series and the T-test method. The global mean value is also provided in the lower right of each panel. See Figure S16 for seasonal plots.

[Figure]

**Figure S14**. Same as Figure S13 but for plant transpiration. See Figure S17 for seasonal plots.

[Figure]

**Figure S15**. Same as Figure S13 but for canopy-intercepted water evaporation. See Figure S18 for seasonal plots.

[Figure]

**Figure S16**. Same as Figure S13 but for seasonal results: (a-d) DJF, (e-h) MAM, (i-l) JJA, and (m-p) SON.

[Figure]

**Figure S17**. Same as Figure S14 but for seasonal results: (a-d) DJF, (e-h) MAM, (i-l) JJA, and (m-p) SON.

[Figure]

**Figure S18**. Same as Figure S15 but for seasonal results: (a-d) DJF, (e-h) MAM, (i-l) JJA, and (m-p) SON.

(5) The updated snow cover parameters can also affect soil moisture particularly in winter, spring, and summer, so we further included some discussions on this aspect in Section 5 as follows:

"*The slightly higher soil moisture in LIS/Noah-MPv5.0 compared to LIS/Noah-MPv4.0.1 is also partially contributed by the updated snow cover parameters in LIS/Noah-MPv5.0 that lead to enhanced snow melting and hence increased soil moisture in winter, spring, and early summer.*"

3. Given the above discrepancy between the soil moisture and latent heat changes, I would also suggest including an additional ET product in the evaluation. While the GLEAM product is certainly a good choice, it is somewhat dependent on the soil moisture assumptions that it makes. So, I would suggest including a soil-moisture independent product like ALEXI to further investigate the conflicting SM and LH responses.

Response: Thank you for the suggestion. We have included the comparison with another soil-moisture independent ET product (FLUXCOM-X-BASE; Nelson et al., 2024) developed based on global FLUXNET flux measurements. We found that the data difference between GLEAM and FLUXCOM is large so that the seemingly conflicting SM and LH biases in the model simulations could be caused by the ET data product uncertainty. This is consistent with previous studies (e.g., Abolafia-Rosenzweig et al., 2021) finding large disagreements across remote sensing-based ET reference datasets. We have added discussions on the observational uncertainties in ET data products in Section 5 as follows:

*"To quantify the uncertainty of reference ET data products, we conduct additional model evaluations using the FLUXCOM-X-BASE (Nelson et al., 2024) data. The results indicate large inconsistency between the FLUXCOM-X-BASE and GLEAM data, where the model biases reverse the signs across many global regions particularly in the low-latitudes (Figure S9). For the CONUS, the bias sign also reverses in the northeastern U.S. and many parts of the western U.S. (Figure S10). Previous studies (Nelson et al., 2024) showed that FLUXCOM-X-BASE has consistently lower ET in evergreen tropics as well as the temperate and high latitudes of the Northern Hemisphere than GLEAM, whereas FLUXCOM-X-BASE has higher ET in the semiarid and arid ecosystems of the lower and middle latitudes. This is consistent with previous studies (e.g., Abolafia-Rosenzweig et al., 2021) finding large disagreements across remote sensing-based ET reference datasets in general. These results suggest that the modeled ET in this study falls into the range of observational uncertainty over many global regions and the uncertainty in ET reference data products can confound model assessments which should be accounted for in future studies."*

References: Nelson, J. A., et al.: X-BASE: the first terrestrial carbon and water flux products from an extended data-driven scaling framework, FLUXCOM-X, Biogeosciences, 21, 5079–5115, https://doi.org/10.5194/bg-21-5079-2024, 2024.

[Figure]

**Figure S9**. Latent heat flux (W/m²) comparison between the GLEAM data, FLUXCOM-X-BASE data, and LIS/Noah-MP simulations driven by USAF forcing globally averaged during 2018-2021: (a) GLEAM3.8a data, (b) LIS/Noah-MPv4.0.1 simulation, (c) LIS/Noah-MPv5.0 simulation, (d) LIS/Noah-MPv4.0.1 biases (model minus GLEAM), (e) LIS/Noah-MPv5.0 biases (model minus GLEAM), (f) differences between LIS/Noah-MPv5.0 and LIS/Noah-MPv4.0.1 simulations, (g) FLUXCOM-X-BASE data, (h) LIS/Noah-MPv4.0.1 biases (model minus FLUXCOM-X-BASE), and (i) LIS/Noah-MPv5.0 biases (model minus FLUXCOM-X-BASE). Grids with statistically significant differences ($p < 0.05$) are shown with gray dots in panels (d)-(f). The statistical significance over each grid is computed using daily time series and the T-test method. The global mean value is also provided in the lower right of each panel.

[Figure]

**Figure S10**. Same as Figure S9 but for evaluation of LIS/Noah-MP simulations driven by the NLDAS-2 forcing over the CONUS.

4. It is a bit unclear how the variables that are evaluated were chosen. From a LIS user perspective, I am wondering whether it would be useful to include an additional figure that would show the results from a more comprehensive and standardized benchmarking framework, like ILAMB, as this would provide a high-level overview of the changes across additional model variables.

Response: We agree with the reviewer that it will be useful to add a summary figure based on ILAMB evaluations to provide a high-level overview of model performance changes. Thus, we have added evaluations using ILAMB and our reference datasets. In addition, we have also added several tables to summarize all the evaluation metrics for all the variables across different global latitudinal bands and seasons. We have included additional discussions in Section 5 as follows:
     *"To summarize the evaluation metrics for all the key variables from both model simulations in this study, we have adopted the International Land Model Benchmarking (ILAMBv2.7.2; Collier et al., 2018) package and applied it to our model simulations and reference datasets. Overall, the result (Figure 18) shows that LIS/Noah-MPv5.0 outperforms or is similar to LIS/Noah-MPv4.0.1 globally in the key land surface variables evaluated in this study, except for slight degradation in simulated surface soil moisture and SWE. In addition, we summarize all the bias values for different seasons and regions for both model simulations in Tables 1 and 2. The slightly degraded surface soil moisture simulation in LIS/Noah-MPv5.0 mainly comes from the degraded performance over northern and southern mid-latitudes, while the slightly degraded SWE in LIS/Noah-MPv5.0 is mainly caused by the degraded performance in the northern high-latitudes (Table 1). The soil moisture and SWE differences between the two model simulations are primarily caused by the model updates in vegetation processes (added canopy heat storage and bug fix of vegetation fraction scaling) and improved snow cover parameters."*

[Figure]

**Figure 18.** Scorecard-type comparison for LIS/Noah-MPv4.0.1 and LIS/Noah-MPv5.0 model performance in simulating key surface variables evaluated against the reference datasets used in this study based on the ILAMB tool.

**Table 1.** Model evaluation metrics for LIS/Noah-MPv4.0.1 and LIS/Noah-MPv5.0 simulations driven by the USAF forcing averaged during 2018-2022 on the global and regional scale. The values are the annual mean model bias (LIS/Noah-MP simulations minus reference datasets). The statistically significant difference between LIS/Noah-MP v4.0.1 and LIS/Noah-MPv5.0 simulations ($p < 0.05$ using a t-test for daily time series) are marked as bold font. The values in the parentheses are the annual mean absolute model biases. The seasonal biases are shown in Tables S1-S4.

| | Global | low latitude (30°S - 30°N) | northern mid-latitudes (30 - 60°N) | northern high-latitudes (>60°N) | southern mid-latitudes (30 - 60°S) | southern high-latitudes (>60°S) |
|---|---|---|---|---|---|---|

| LIS/Noah-MP | v4.0.1 | v5.0 | v4.0.1 | v5.0 | v4.0.1 | v5.0 | v4.0.1 | v5.0 | v4.0.1 | v5.0 | v4.0.1 | v5.0 |
|---|---|---|---|---|---|---|---|---|---|---|---|---|
| Surface soil moisture ($m^3/m^3$ compared to SMAP) | **0.003 (0.076)** | **0.008 (0.078)** | **-0.009 (0.065)** | **-0.002 (0.066)** | **0.020 (0.079)** | **0.025 (0.082)** | **-0.013 (0.093)** | **-0.009 (0.094)** | **0.028 (0.081)** | **0.036 (0.086)** | - | - |
| Surface Soil moisture ($m^3/m^3$ compared to ISMN) | **0.062 (0.078)** | **0.067 (0.082)** | **0.027 (0.061)** | **0.036 (0.067)** | **0.062 (0.079)** | **0.068 (0.082)** | 0.119 (0.121) | 0.121 (0.123) | 0.049 (0.062) | 0.051 (0.062) | - | - |
| Latent heat flux ($W/m^2$ compared to GLEAM3.8a) | **0.992 (6.802)** | **-0.386 (7.273)** | **2.105 (10.740)** | **-2.759 (11.601)** | 0.752 (7.994) | -0.608 (8.127) | -4.122 (5.541) | -3.784 (5.731) | **1.469 (9.369)** | **-0.271 (9.627)** | **2.992 (3.105)** | **3.668 (3.692)** |
| Snow water equivalent (mm compared to ERA5-Land) | **-10.123 (22.444)** | **-13.237 (22.328)** | **-0.845 (0.951)** | **-0.878 (0.966)** | **0.715 (16.267)** | **-1.349 (15.898)** | **-45.177 (71.928)** | **-56.181 (72.276)** | -10.804 (16.494) | -10.471 (16.311) | - | - |
| Snow depth (m compared to ERA5-Land) | -0.059 (0.076) | -0.061 (0.079) | **-0.003 (0.003)** | **-0.003 (0.003)** | -0.019 (0.051) | -0.019 (0.052) | -0.231 (0.255) | -0.245 (0.268) | -0.040 (0.050) | -0.037 (0.050) | - | - |
| Snow cover fraction (compared to MODIS) | **0.112 (0.113)** | **0.069 (0.090)** | **0.001 (0.003)** | **0.000 (0.002)** | **0.149 (0.151)** | **0.118 (0.122)** | **0.234 (0.235)** | **0.108 (0.183)** | **0.020 (0.027)** | **0.015 (0.023)** | - | - |
| Surface albedo (compared to MODIS) | **-0.018 (0.061)** | **-0.033 (0.067)** | **-0.016 (0.047)** | **-0.017 (0.046)** | **0.032 (0.052)** | **0.021 (0.045)** | **0.016 (0.052)** | **-0.024 (0.072)** | **0.017 (0.034)** | **0.013 (0.032)** | **-0.084 (0.089)** | **-0.100 (0.102)** |

**Table 2.** Model evaluation metrics for LIS/Noah-MPv4.0.1 and LIS/Noah-MPv5.0 simulations driven by the NLDAS-2 forcing averaged over the CONUS during 2018-2022. The values are the mean model bias (LIS/Noah-MP simulations minus reference datasets). The statistically significant difference between LIS/Noah-MP v4.0.1 and LIS/Noah-MPv5.0 simulations ($p < 0.05$ using a t-test for daily time series) are marked as bold font. The values in the parentheses are the mean absolute model biases.

| | Annual | | DJF | | MAM | | JJA | | SON | |
|---|---|---|---|---|---|---|---|---|---|---|
| LIS/Noah-MP | v4.0.1 | v5.0 | v4.0.1 | v5.0 | v4.0.1 | v5.0 | v4.0.1 | v5.0 | v4.0.1 | v5.0 |
| Surface soil moisture ($m^3/m^3$ compared to SMAP) | **0.000 (0.062)** | **0.008 (0.065)** | **0.025 (0.077)** | **0.035 (0.085)** | **0.003 (0.067)** | **0.008 (0.069)** | **0.006 (0.062)** | **0.013 (0.065)** | **-0.010 (0.058)** | **-0.001 (0.062)** |
| Surface Soil moisture ($m^3/m^3$ compared to ISMN) | **0.041 (0.065)** | **0.047 (0.068)** | **0.041 (0.075)** | **0.051 (0.080)** | **0.024 (0.066)** | **0.029 (0.067)** | **0.043 (0.069)** | **0.049 (0.072)** | **0.047 (0.069)** | **0.054 (0.074)** |
| Latent heat flux ($W/m^2$ compared to GLEAM3.8a) | -0.207 (9.135) | -2.302 (9.286) | -5.864 (7.014) | -5.126 (6.385) | -0.575 (14.912) | -3.498 (14.413) | 9.476 (17.752) | 3.209 (14.815) | -4.017 (7.147) | -3.865 (7.904) |

| | | | | | | | | | | |
|---|---|---|---|---|---|---|---|---|---|---|
| Snow water equivalent (mm compared to SNODAS) | **-4.173 (6.422)** | **-4.959 (6.369)** | **-5.083 (10.148)** | **-6.715 (9.961)** | -10.246 (13.924) | -11.309 (14.061) | **-0.700 (1.221)** | **-0.961 (1.051)** | **-0.643 (1.018)** | **-0.843 (0.930)** |
| Snow depth (m compared to SNODAS) | -0.013 (0.020) | -0.015 (0.020) | -0.016 (0.036) | -0.020 (0.035) | -0.032 (0.040) | -0.033 (0.040) | **-0.002 (0.003)** | **-0.002 (0.002)** | -0.004 (0.005) | -0.004 (0.005) |
| Snow cover fraction (compared to MODIS) | **0.055 (0.058)** | **0.028 (0.037)** | **0.221 (0.227)** | **0.117 (0.137)** | **0.045 (0.049)** | **0.026 (0.046)** | -0.003 (0.003) | -0.003 (0.003) | **0.018 (0.026)** | **0.004 (0.018)** |
| Surface albedo (compared to MODIS) | **0.031 (0.038)** | **0.023 (0.033)** | **0.072 (0.083)** | **0.030 (0.056)** | **0.022 (0.032)** | **0.016 (0.029)** | **0.024 (0.033)** | **0.023 (0.033)** | **0.031 (0.041)** | **0.026 (0.037)** |

Minor comments:

1. Generally, a lot of the Figures are small, making it hard to see details. Since several panels often share a color bar, maybe you can just have the two color bars at the side, which might allow you to increase the subpanel size.

Response: Thank you for the suggestion. We have revised all the figures in the main text and Figures S2, S4, S5, and S8 in the supplement by increasing subpanel sizes and font sizes. Please see the track-change revised manuscript for this change.

2. Line 24: Here and elsewhere in the paper I would suggest avoiding formulations like 'negative bias' and instead say 'under-/overestimates soil moisture', as this is more intuitive.

Response: As suggested, we have fixed this issue throughout the manuscript. Please see the track-change revised manuscript.

3. Lines 104 – 105: If a 'scorecard' type evaluation is the intention, then I don't understand why the authors opted not to use one of the existing comprehensive benchmarking tools, like ILAMB (see major comment) or include a scorecard type figure in the paper. Also, the quotation marks are mismatched.

Response: As suggested, we have included additional evaluation and figure using the ILAMB tool. We have included additional discussions related to this in Section 5 (see also our response to your major comment #4 above for details) and also fixed the quotation marks. The scorecard type of figure based on the ILAMB tool is as follows:

[Figure]

**Figure 18.** Scorecard-type comparison for LIS/Noah-MPv4.0.1 and LIS/Noah-MPv5.0 model performance in simulating key surface variables evaluated against the reference datasets used in this study based on the ILAMB tool.

4. Line 147: "…control the soil process timestep."

Response: Thanks. We fixed this issue as suggested.

5. Line 159: "…of benchmark simulations with LIS coupled with Noah-MP". Rephrasing to avoid 'coupled simulations' since it implies coupled to a GCM.

Response: Thanks. We fixed this issue as suggested.

6. Lines 174 – 175: Was STATSGO used over the US and FAO elsewhere?

Response: Yes, the STATSGO soil data was used for the U.S. and the FAO soil data was used for other global regions outside the U.S.

7. Lines 218 – 219: There are several areas where the precipitation bias does not align with the soil moisture bias (for example Northern Canada or Southern Brazil). What is the suspected cause for the soil moisture differences there?

Response: Thanks for the great question. At this stage, we do not have a definite answer for this question. Several possible causes may play a role, including observational data uncertainties (i.e., IMERG/GPM precipitation and SMAP soil moisture), model ET biases induced by uncertainties in other meteorological forcing and/or model soil hydraulics, and soil infiltration parameterization uncertainty in the model. This requires further investigations. We have included the following sentence in Lines 234-235 (track-changed revised manuscript) for further clarification:
      "…, *partially caused by the USAF precipitation forcing bias (Figure S2), except for northern Canada and southern Brazil which requires further investigation*."

8. Figure 2: Here and in all other bias figures, I think it would be helpful to include the mean absolute bias values as well, to get a sense for the model changes without the impact of compensating errors.

Response: Thanks for the suggestion. Instead of adding the mean absolute bias value in the figure panels, we have created a table (see Tables 1 and 2 below) to summarize the model mean bias and mean absolute bias. We have added these tables and related discussions in Section 5 of the revised manuscript.

**Table 1.** Model evaluation metrics for LIS/Noah-MPv4.0.1 and LIS/Noah-MPv5.0 simulations driven by the USAF forcing averaged during 2018-2022 on the global and regional scale. The values are the annual mean model bias (LIS/Noah-MP simulations minus reference datasets). The statistically significant difference between LIS/Noah-MP v4.0.1 and LIS/Noah-MPv5.0 simulations ($p < 0.05$ using a t-test for daily time series) are marked as bold font. The values in the parentheses are the annual mean absolute model biases. The seasonal biases are shown in Tables S1-S4.

| | Global | | low latitude (30°S - 30°N) | | northern mid-latitudes (30 - 60°N) | | northern high-latitudes (>60°N) | | southern mid-latitudes (30 - 60°S) | | southern high-latitudes (>60°S) | |
|---|---|---|---|---|---|---|---|---|---|---|---|---|
| LIS/Noah-MP | v4.0.1 | v5.0 | v4.0.1 | v5.0 | v4.0.1 | v5.0 | v4.0.1 | v5.0 | v4.0.1 | v5.0 | v4.0.1 | v5.0 |
| Surface soil moisture ($m^3/m^3$ compared to SMAP) | **0.003 (0.076)** | **0.008 (0.078)** | **-0.009 (0.065)** | **-0.002 (0.066)** | **0.020 (0.079)** | **0.025 (0.082)** | **-0.013 (0.093)** | **-0.009 (0.094)** | **0.028 (0.081)** | **0.036 (0.086)** | - | - |

| | | | | | | | | | | | |
|---|---|---|---|---|---|---|---|---|---|---|---|
| Surface Soil moisture (m³/m³ compared to ISMN) | **0.062 (0.078)** | **0.067 (0.082)** | **0.027 (0.061)** | **0.036 (0.067)** | **0.062 (0.079)** | **0.068 (0.082)** | 0.119 (0.121) | 0.121 (0.123) | 0.049 (0.062) | 0.051 (0.062) | - | - |
| Latent heat flux (W/m² compared to GLEAM3.8a) | **0.992 (6.802)** | **-0.386 (7.273)** | **2.105 (10.740)** | **-2.759 (11.601)** | 0.752 (7.994) | -0.608 (8.127) | -4.122 (5.541) | -3.784 (5.731) | **1.469 (9.369)** | **-0.271 (9.627)** | **2.992 (3.105)** | **3.668 (3.692)** |
| Snow water equivalent (mm compared to ERA5-Land) | **-10.123 (22.444)** | **-13.237 (22.328)** | **-0.845 (0.951)** | **-0.878 (0.966)** | **0.715 (16.267)** | **-1.349 (15.898)** | **-45.177 (71.928)** | **-56.181 (72.276)** | -10.804 (16.494) | -10.471 (16.311) | - | - |
| Snow depth (m compared to ERA5-Land) | -0.059 (0.076) | -0.061 (0.079) | **-0.003 (0.003)** | **-0.003 (0.003)** | -0.019 (0.051) | -0.019 (0.052) | -0.231 (0.255) | -0.245 (0.268) | -0.040 (0.050) | -0.037 (0.050) | - | - |
| Snow cover fraction (compared to MODIS) | **0.112 (0.113)** | **0.069 (0.090)** | **0.001 (0.003)** | **0.000 (0.002)** | **0.149 (0.151)** | **0.118 (0.122)** | **0.234 (0.235)** | **0.108 (0.183)** | **0.020 (0.027)** | **0.015 (0.023)** | - | - |
| Surface albedo (compared to MODIS) | **-0.018 (0.061)** | **-0.033 (0.067)** | **-0.016 (0.047)** | **-0.017 (0.046)** | **0.032 (0.052)** | **0.021 (0.045)** | **0.016 (0.052)** | **-0.024 (0.072)** | **0.017 (0.034)** | **0.013 (0.032)** | **-0.084 (0.089)** | **-0.100 (0.102)** |

**Table 2.** Model evaluation metrics for LIS/Noah-MPv4.0.1 and LIS/Noah-MPv5.0 simulations driven by the NLDAS-2 forcing averaged over the CONUS during 2018-2022. The values are the mean model bias (LIS/Noah-MP simulations minus reference datasets). The statistically significant difference between LIS/Noah-MP v4.0.1 and LIS/Noah-MPv5.0 simulations ($p < 0.05$ using a t-test for daily time series) are marked as bold font. The values in the parentheses are the mean absolute model biases.

| | Annual | | DJF | | MAM | | JJA | | SON | |
|---|---|---|---|---|---|---|---|---|---|---|
| LIS/Noah-MP | v4.0.1 | v5.0 | v4.0.1 | v5.0 | v4.0.1 | v5.0 | v4.0.1 | v5.0 | v4.0.1 | v5.0 |
| Surface soil moisture (m³/m³ compared to SMAP) | **0.000 (0.062)** | **0.008 (0.065)** | **0.025 (0.077)** | **0.035 (0.085)** | **0.003 (0.067)** | **0.008 (0.069)** | **0.006 (0.062)** | **0.013 (0.065)** | **-0.010 (0.058)** | **-0.001 (0.062)** |
| Surface Soil moisture (m³/m³ compared to ISMN) | **0.041 (0.065)** | **0.047 (0.068)** | **0.041 (0.075)** | **0.051 (0.080)** | **0.024 (0.066)** | **0.029 (0.067)** | **0.043 (0.069)** | **0.049 (0.072)** | **0.047 (0.069)** | **0.054 (0.074)** |
| Latent heat flux (W/m² compared to GLEAM3.8a) | **-0.207 (9.135)** | **-2.302 (9.286)** | **-5.864 (7.014)** | **-5.126 (6.385)** | **-0.575 (14.912)** | **-3.498 (14.413)** | **9.476 (17.752)** | **3.209 (14.815)** | -4.017 (7.147) | -3.865 (7.904) |
| Snow water equivalent (mm compared to SNODAS) | **-4.173 (6.422)** | **-4.959 (6.369)** | **-5.083 (10.148)** | **-6.715 (9.961)** | -10.246 (13.924) | -11.309 (14.061) | **-0.700 (1.221)** | **-0.961 (1.051)** | **-0.643 (1.018)** | **-0.843 (0.930)** |

| | | | | | | | | | | |
|---|---|---|---|---|---|---|---|---|---|---|
| Snow depth (m compared to SNODAS) | -0.013 (0.020) | -0.015 (0.020) | -0.016 (0.036) | -0.020 (0.035) | -0.032 (0.040) | -0.033 (0.040) | **-0.002 (0.003)** | **-0.002 (0.002)** | -0.004 (0.005) | -0.004 (0.005) |
| Snow cover fraction (compared to MODIS) | **0.055 (0.058)** | **0.028 (0.037)** | **0.221 (0.227)** | **0.117 (0.137)** | **0.045 (0.049)** | **0.026 (0.046)** | -0.003 (0.003) | -0.003 (0.003) | **0.018 (0.026)** | **0.004 (0.018)** |
| Surface albedo (compared to MODIS) | **0.031 (0.038)** | **0.023 (0.033)** | **0.072 (0.083)** | **0.030 (0.056)** | **0.022 (0.032)** | **0.016 (0.029)** | **0.024 (0.033)** | **0.023 (0.033)** | **0.031 (0.041)** | **0.026 (0.037)** |

9. Lines 314 – 315: "…uses snowpack physics consistent with other land snowpacks…"

Response: Thanks. We fixed this issue as suggested.

10. Line 329: "…despite the overestimation of soil moisture…"

Response: Thanks. We fixed this issue as suggested.

11. Line 418 – 419: What do the authors think is the reason that these new parameters are only effective at reducing the bias in some regions?

Response: Thanks for the question. There are three possible reasons. (1) The snow cover parameter updates are more effective for regions with snow depth less than about 0.3 m, since this is the most sensitive snow depth regime for snow cover calculations based on the parameterization used in Noah-MP (He et al, 2019). (2) The snow cover parameter updates are also vegetation type dependent, so the effectiveness of this change also depends on vegetation types, leading to regional heterogeneity in the results. (3) Due to the positive surface albedo feedback induced by snow cover change, it is more effective over ablation regions and periods. (4) The snow cover impact is further complicated by the spatial heterogeneity of SWE biases (Abolafia-Rosenzweig et al., 2025). We have included these discussions in the track-changed revised manuscript (Lines 473-483).

---

## Author Comment (AC2)

**Reviewer #2 (Vincent Fortin):**

This paper presents a detailed evaluation of an upgrade to the NASA LIS. The paper is well written and should be useful to LIS users.

Response: We thank the reviewer for the positive feedback and the constructive comments, which help to improve our manuscript quality. We have addressed all the comments point by point below.

Comments:

1. When summarizing the results, I suggest adding a table, in the form of a scorecard (with a stratification per variable, season and domains), that summarizes the magnitude and significance of the changes in the results obtained for the two LIS versions that are evaluated in the paper.

Response: Thank you for the suggestions. We have added multiple tables to summarize model biases for each of the evaluated variables across different seasons and regions, with both the bias magnitude and significance of the resulting changes included in the tables. In addition, we also included a scorecard-type figure using the ILAMB evaluation tool. These tables and figures and relevant discussions have been added to Section 5 of the revised manuscript and supplement. Below are the added figure and tables.

**Table 1.** Model evaluation metrics for LIS/Noah-MPv4.0.1 and LIS/Noah-MPv5.0 simulations driven by the USAF forcing averaged during 2018-2022 on the global and regional scale. The values are the annual mean model bias (LIS/Noah-MP simulations minus reference datasets). The statistically significant difference between LIS/Noah-MP v4.0.1 and LIS/Noah-MPv5.0 simulations ($p < 0.05$ using a t-test for daily time series) are marked as bold font. The values in the parentheses are the mean absolute model biases. The seasonal biases are shown in Tables S1-S4.

| | Global | | low latitude (30°S - 30°N) | | northern mid-latitudes (30 - 60°N) | | northern high-latitudes (>60°N) | | southern mid-latitudes (30 - 60°S) | | southern high-latitudes (>60°S) | |
|---|---|---|---|---|---|---|---|---|---|---|---|---|
| LIS/Noah-MP | v4.0.1 | v5.0 | v4.0.1 | v5.0 | v4.0.1 | v5.0 | v4.0.1 | v5.0 | v4.0.1 | v5.0 | v4.0.1 | v5.0 |
| Surface soil moisture ($m^3/m^3$ compared to SMAP) | **0.003 (0.076)** | **0.008 (0.078)** | **-0.009 (0.065)** | **-0.002 (0.066)** | **0.020 (0.079)** | **0.025 (0.082)** | **-0.013 (0.093)** | **-0.009 (0.094)** | **0.028 (0.081)** | **0.036 (0.086)** | - | - |
| Surface Soil moisture ($m^3/m^3$ compared to ISMN) | **0.062 (0.078)** | **0.067 (0.082)** | **0.027 (0.061)** | **0.036 (0.067)** | **0.062 (0.079)** | **0.068 (0.082)** | 0.119 (0.121) | 0.121 (0.123) | 0.049 (0.062) | 0.051 (0.062) | - | - |
| Latent heat flux ($W/m^2$ compared to GLEAM3.8a) | **0.992 (6.802)** | **-0.386 (7.273)** | **2.105 (10.740)** | **-2.759 (11.601)** | 0.752 (7.994) | -0.608 (8.127) | -4.122 (5.541) | -3.784 (5.731) | **1.469 (9.369)** | **-0.271 (9.627)** | **2.992 (3.105)** | **3.668 (3.692)** |
| Snow water equivalent (mm compared to ERA5-Land) | **-10.123 (22.444)** | **-13.237 (22.328)** | **-0.845 (0.951)** | **-0.878 (0.966)** | **0.715 (16.267)** | **-1.349 (15.898)** | **-45.177 (71.928)** | **-56.181 (72.276)** | -10.804 (16.494) | -10.471 (16.311) | - | - |

| Snow depth (m compared to ERA5-Land) | -0.059 (0.076) | -0.061 (0.079) | **-0.003 (0.003)** | **-0.003 (0.003)** | -0.019 (0.051) | -0.019 (0.052) | -0.231 (0.255) | -0.245 (0.268) | -0.040 (0.050) | -0.037 (0.050) | - | - |
|---|---|---|---|---|---|---|---|---|---|---|---|---|
| Snow cover fraction (compared to MODIS) | **0.112 (0.113)** | **0.069 (0.090)** | **0.001 (0.003)** | **0.000 (0.002)** | **0.149 (0.151)** | **0.118 (0.122)** | **0.234 (0.235)** | **0.108 (0.183)** | **0.020 (0.027)** | **0.015 (0.023)** | - | - |
| Surface albedo (compared to MODIS) | -0.018 (0.061) | -0.033 (0.067) | -0.016 (0.047) | -0.017 (0.046) | 0.032 (0.052) | 0.021 (0.045) | 0.016 (0.052) | -0.024 (0.072) | 0.017 (0.034) | 0.013 (0.032) | -0.084 (0.089) | -0.100 (0.102) |

**Table 2.** Model evaluation metrics for LIS/Noah-MPv4.0.1 and LIS/Noah-MPv5.0 simulations driven by the NLDAS-2 forcing averaged over the CONUS during 2018-2022. The values are the mean model bias (LIS/Noah-MP simulations minus reference datasets). The statistically significant difference between LIS/Noah-MP v4.0.1 and LIS/Noah-MPv5.0 simulations ($p < 0.05$ using a t-test for daily time series) are marked as bold font. The values in the parentheses are the mean absolute model biases.

| | Annual | | DJF | | MAM | | JJA | | SON | |
|---|---|---|---|---|---|---|---|---|---|---|
| LIS/Noah-MP | v4.0.1 | v5.0 | v4.0.1 | v5.0 | v4.0.1 | v5.0 | v4.0.1 | v5.0 | v4.0.1 | v5.0 |
| Surface soil moisture ($m^3/m^3$ compared to SMAP) | **0.000 (0.062)** | **0.008 (0.065)** | **0.025 (0.077)** | **0.035 (0.085)** | **0.003 (0.067)** | **0.008 (0.069)** | **0.006 (0.062)** | **0.013 (0.065)** | **-0.010 (0.058)** | **-0.001 (0.062)** |
| Surface Soil moisture ($m^3/m^3$ compared to ISMN) | **0.041 (0.065)** | **0.047 (0.068)** | **0.041 (0.075)** | **0.051 (0.080)** | **0.024 (0.066)** | **0.029 (0.067)** | **0.043 (0.069)** | **0.049 (0.072)** | **0.047 (0.069)** | **0.054 (0.074)** |
| Latent heat flux ($W/m^2$ compared to GLEAM3.8a) | **-0.207 (9.135)** | **-2.302 (9.286)** | **-5.864 (7.014)** | **-5.126 (6.385)** | **-0.575 (14.912)** | **-3.498 (14.413)** | **9.476 (17.752)** | **3.209 (14.815)** | -4.017 (7.147) | -3.865 (7.904) |
| Snow water equivalent (mm compared to SNODAS) | **-4.173 (6.422)** | **-4.959 (6.369)** | **-5.083 (10.148)** | **-6.715 (9.961)** | -10.246 (13.924) | -11.309 (14.061) | **-0.700 (1.221)** | **-0.961 (1.051)** | **-0.643 (1.018)** | **-0.843 (0.930)** |
| Snow depth (m compared to SNODAS) | -0.013 (0.020) | -0.015 (0.020) | -0.016 (0.036) | -0.020 (0.035) | -0.032 (0.040) | -0.033 (0.040) | **-0.002 (0.003)** | **-0.002 (0.002)** | -0.004 (0.005) | -0.004 (0.005) |
| Snow cover fraction (compared to MODIS) | **0.055 (0.058)** | **0.028 (0.037)** | **0.221 (0.227)** | **0.117 (0.137)** | **0.045 (0.049)** | **0.026 (0.046)** | -0.003 (0.003) | -0.003 (0.003) | **0.018 (0.026)** | **0.004 (0.018)** |
| Surface albedo (compared to MODIS) | **0.031 (0.038)** | **0.023 (0.033)** | **0.072 (0.083)** | **0.030 (0.056)** | **0.022 (0.032)** | **0.016 (0.029)** | **0.024 (0.033)** | **0.023 (0.033)** | **0.031 (0.041)** | **0.026 (0.037)** |

**Table S1.** Model evaluation metrics for LIS/Noah-MPv4.0.1 and LIS/Noah-MPv5.0 simulations driven by the USAF forcing averaged over December-January-February (DJF) during 2018-2022 on the global and regional scale. The values are the mean model bias (LIS/Noah-MP simulations minus reference datasets). The statistically significant difference between LIS/Noah-MP v4.0.1 and LIS/Noah-MPv5.0 simulations ($p < 0.05$ using a t-test for daily time series) are marked as bold font. The values in the parentheses are the mean absolute model biases.

| | Global | | Low latitude | | Northern midlatitude | | Northern high latitude | | Southern midlatitude | | Southern high latitude | |
|---|---|---|---|---|---|---|---|---|---|---|---|---|
| LIS/Noah-MP | v4.0.1 | v5.0 | v4.0.1 | v5.0 | v4.0.1 | v5.0 | v4.0.1 | v5.0 | v4.0.1 | v5.0 | v4.0.1 | v5.0 |
| Surface soil moisture ($m^3/m^3$ compared to SMAP) | **0.020 (0.080)** | **0.027 (0.083)** | **-0.004 (0.070)** | **0.004 (0.072)** | **0.059 (0.095)** | **0.066 (0.100)** | -0.063 (0.158) | -0.062 (0.157) | **0.035 (0.082)** | **0.043 (0.087)** | - | - |
| Surface Soil moisture ($m^3/m^3$ compared to ISMN) | **0.052 (0.078)** | **0.059 (0.084)** | **0.013 (0.053)** | **0.022 (0.059)** | **0.055 (0.081)** | **0.062 (0.086)** | 0.096 (0.099) | 0.096 (0.099) | 0.065 (0.068) | 0.068 (0.070) | - | - |
| Latent heat flux ($W/m^2$ compared to GLEAM3.8a) | **2.414 (8.184)** | **1.466 (8.297)** | **2.141 (16.165)** | **-3.752 (14.870)** | **-2.962 (4.229)** | **-2.449 (4.016)** | **-0.113 (0.894)** | **-0.190 (0.916)** | **12.486 (21.025)** | **6.452 (18.489)** | **7.447 (7.659)** | **9.338 (9.442)** |
| Snow water equivalent (mm compared to ERA5-Land) | **-6.051 (26.573)** | **-7.823 (26.321)** | **-0.853 (0.984)** | **-0.883 (1.000)** | 7.218 (23.983) | 5.418 (23.245) | **-37.503 (78.639)** | **-42.473 (79.038)** | -8.938 (12.807) | -8.728 (12.370) | - | - |
| Snow depth (m compared to ERA5-Land) | **-0.073 (0.102)** | **-0.066 (0.100)** | -0.003 (0.004) | -0.004 (0.004) | **-0.023 (0.083)** | **-0.015 (0.083)** | **-0.293 (0.324)** | **-0.272 (0.315)** | -0.031 (0.039) | -0.030 (0.037) | - | - |
| Snow cover fraction (compared to MODIS) | **0.266 (0.267)** | **0.202 (0.206)** | **0.004 (0.006)** | **0.003 (0.004)** | **0.386 (0.389)** | **0.311 (0.317)** | **0.516 (0.516)** | **0.366 (0.373)** | **0.005 (0.009)** | **0.004 (0.009)** | - | - |
| Surface albedo (compared to MODIS) | **-0.009 (0.091)** | **-0.031 (0.093)** | **-0.018 (0.051)** | **-0.020 (0.051)** | **0.094 (0.130)** | **0.056 (0.102)** | **0.005 (0.121)** | **-0.053 (0.172)** | **0.011 (0.033)** | **0.009 (0.032)** | **-0.086 (0.090)** | **-0.101 (0.103)** |

**Table S2.** Same as Table S1 but for March-April-May (MAM) averages.

| | Global | | Low latitude | | Northern midlatitude | | Northern high latitude | | Southern midlatitude | | Southern high latitude | |
|---|---|---|---|---|---|---|---|---|---|---|---|---|
| LIS/Noah-MP | v4.0.1 | v5.0 | v4.0.1 | v5.0 | v4.0.1 | v5.0 | v4.0.1 | v5.0 | v4.0.1 | v5.0 | v4.0.1 | v5.0 |
| Surface soil moisture ($m^3/m^3$ compared to SMAP) | **0.013 (0.077)** | **0.016 (0.078)** | **-0.006 (0.069)** | **0.000 (0.070)** | **0.033 (0.082)** | **0.036 (0.084)** | 0.009 (0.081)) | 0.008 (0.081) | **0.021 (0.077)** | **0.030 (0.083)** | - | - |

| | Global | | Low latitude | | Northern midlatitude | | Northern high latitude | | Southern midlatitude | | Southern high latitude | |
|---|---|---|---|---|---|---|---|---|---|---|---|---|
| Surface Soil moisture (m³/m³ compared to ISMN) | **0.048 (0.075)** | **0.051 (0.077)** | **0.032 (0.063)** | **0.040 (0.066)** | **0.048 (0.075)** | **0.051 (0.077)** | 0.106 (0.116) | 0.107 (0.117) | 0.054 (0.060) | 0.057 (0.058) | - | - |
| Latent heat flux (W/m² compared to GLEAM3.8a) | **-1.614 (8.272)** | **-2.609 (8.499)** | **-0.028 (13.243)** | **-4.268 (14.196)** | -2.284 (12.144) | -3.301 (12.521) | **-6.919 (8.214)** | **-5.205 (7.419)** | -1.425 (6.665) | -0.836 (7.569) | **0.448 (1.295)** | **0.557 (1.306)** |
| Snow water equivalent (mm compared to ERA5-Land) | **-8.642 (29.319)** | **-15.069 (30.461)** | **-1.064 (1.272)** | **-1.116 (1.291)** | **2.464 (26.332)** | **-1.265 (27.002)** | **-41.038 (87.507)** | **-65.033 (92.067)** | **-9.468 (11.037)** | **-9.000 (10.882)** | | |
| Snow depth (m compared to ERA5-Land) | **-0.068 (0.096)** | **-0.078 (0.107)** | **-0.004 (0.004)** | **-0.004 (0.004)** | -0.026 (0.075) | -0.027 (0.080) | **-0.265 (0.308)** | **-0.313 (0.354)** | **-0.032 (0.035)** | **-0.031 (0.035)** | - | - |
| Snow cover fraction (compared to MODIS) | **0.129 (0.131)** | **0.064 (0.121)** | **0.001 (0.003)** | **0.000 (0.002)** | **0.160 (0.162)** | **0.126 (0.146)** | **0.291 (0.291)** | **0.075 (0.274)** | **0.005 (0.016)** | **0.002 (0.014)** | - | - |
| Surface albedo (compared to MODIS) | **-0.023 (0.064)** | **-0.048 (0.081)** | **-0.020 (0.048)** | **-0.021 (0.048)** | 0.029 (0.059) | 0.014 (0.056) | -0.054 (0.092) | -0.152 (0.180) | 0.018 (0.036) | 0.015 (0.035) | -0.063 (0.069) | -0.072 (0.075) |

**Table S3.** Same as Table S1 but for June-July-August (JJA) averages.

| | Global | | Low latitude | | Northern midlatitude | | Northern high latitude | | Southern midlatitude | | Southern high latitude | |
|---|---|---|---|---|---|---|---|---|---|---|---|---|
| Noah-MP | v4.0.1 | v5.0 | v4.0.1 | v5.0 | v4.0.1 | v5.0 | v4.0.1 | v5.0 | v4.0.1 | v5.0 | v4.0.1 | v5.0 |
| Surface soil moisture (m³/m³ compared to SMAP) | **-0.005 (0.078)** | **0.000 (0.079)** | **-0.016 (0.071)** | **-0.010 (0.071)** | **0.012 (0.078)** | **0.017 (0.081)** | **-0.023 (0.091)** | **-0.019 (0.091)** | **0.022 (0.090)** | **0.031 (0.096)** | - | - |
| Surface Soil moisture (m³/m³ compared to ISMN) | **0.067 (0.085)** | **0.072 (0.088)** | **0.024 (0.070)** | **0.034 (0.075)** | **0.069 (0.084)** | **0.074 (0.088)** | 0.127 (0.131) | 0.129 (0.132) | 0.039 (0.071) | 0.039 (0.070) | - | - |
| Latent heat flux (W/m² compared to GLEAM3.8a) | **2.927 (11.336)** | **0.472 (10.599)** | **4.066 (17.233)** | **-0.412 (16.601)** | **10.697 (17.801)** | **5.522 (14.889)** | -5.468 (12.165) | -6.020 (12.962) | **-4.440 (7.092)** | **-3.425 (6.825)** | 0.959 (1.286) | 1.012 (1.313) |
| Snow water equivalent (mm compared to ERA5-Land) | **-13.131 (17.021)** | **-15.966 (17.519)** | **-0.740 (0.813)** | **-0.772 (0.821)** | **-3.316 (8.360)** | **-5.265 (7.901)** | **-52.729 (59.916)** | **-62.524 (63.347)** | -10.688 (18.101) | -10.761 (18.218) | - | - |
| Snow depth (m compared to ERA5-Land) | **-0.047 (0.054)** | **-0.053 (0.056)** | **-0.003 (0.003)** | **-0.003 (0.003)** | **-0.014 (0.024)** | **-0.018 (0.023)** | **-0.184 (0.197)** | **-0.206 (0.208)** | -0.043 (0.061) | -0.041 (0.062) | - | - |
| Snow cover fraction (compared to MODIS) | **0.017 (0.030)** | **0.006 (0.034)** | **0.000 (0.002)** | **-0.001 (0.002)** | **0.001 (0.012)** | **-0.003 (0.011)** | **0.060 (0.093)** | **0.024 (0.112)** | **0.057 (0.071)** | **0.039 (0.055)** | - | - |

| Surface albedo (compared to MODIS) | 0.005 (0.045) | -0.001 (0.045) | -0.018 (0.049) | -0.019 (0.049) | 0.011 (0.037) | 0.010 (0.036) | 0.027 (0.050) | 0.009 (0.051) | 0.032 (0.044) | 0.023 (0.038) | -0.036 (0.045) | -0.053 (0.059) |

**Table S4.** Same as Table S1 but for September-October-November (SON) averages.

| | Global | | Low latitude | | Northern midlatitude | | Northern high latitude | | Southern midlatitude | | Southern high latitude | |
|---|---|---|---|---|---|---|---|---|---|---|---|---|
| Noah-MP | v4.0.1 | v5.0 | v4.0.1 | v5.0 | v4.0.1 | v5.0 | v4.0.1 | v5.0 | v4.0.1 | v5.0 | v4.0.1 | v5.0 |
| Surface soil moisture ($m^3/m^3$ compared to SMAP) | 0.008 (0.080) | 0.014 (0.082) | -0.010 (0.065) | -0.004 (0.065) | 0.020 (0.080) | 0.027 (0.084) | 0.014 (0.108) | 0.019 (0.110) | 0.036 (0.090) | 0.042 (0.093) | - | - |
| Surface Soil moisture ($m^3/m^3$ compared to ISMN) | 0.066 (0.083) | 0.073 (0.088) | 0.026 (0.071) | 0.034 (0.074) | 0.067 (0.083) | 0.074 (0.088) | 0.120 (0.120) | 0.121 (0.121) | 0.061 (0.082) | 0.061 (0.079) | - | - |
| Latent heat flux ($W/m^2$ compared to GLEAM3.8a) | 0.301 (6.785) | -0.681 (7.228) | 2.243 (12.533) | -2.624 (13.228) | -2.550 (6.536) | -2.259 (6.910 | -3.766 (4.360) | -3.522 (4.166) | -0.550 (10.947) | -3.173 (11.249) | 3.204 (3.237) | 3.879 (3.883) |
| Snow water equivalent (mm compared to ERA5-Land) | -12.029 (17.559) | -13.400 (17.511) | -0.698 (0.729) | -0.718 (0.743) | -3.228 (7.672) | -3.980 (7.407) | -47.344 (63.135) | -52.270 (63.555) | -12.553 (22.812) | -12.550 (22.978) | - | - |
| Snow depth (m compared to ERA5-Land) | -0.045 (0.060) | -0.046 (0.061) | -0.002 (0.003) | -0.003 (0.003) | -0.014 (0.028) | -0.014 (0.028) | -0.175 (0.215) | -0.180 (0.221) | -0.044 (0.065) | -0.043 (0.065) | - | - |
| Snow cover fraction (compared to MODIS) | 0.114 (0.116) | 0.077 (0.082) | 0.000 (0.001) | -0.001 (0.001) | 0.111 (0.116) | 0.084 (0.090) | 0.297 (0.298) | 0.189 (0.195) | 0.023 (0.029) | 0.020 (0.027) | - | - |
| Surface albedo (compared to MODIS) | -0.012 (0.065) | -0.025 (0.063) | -0.018 (0.046) | -0.019 (0.046) | 0.034 (0.056) | 0.025 (0.048) | 0.057 (0.066) | 0.022 (0.047) | 0.013 (0.031) | 0.010 (0.029) | -0.088 (0.091) | -0.102 (0.103) |

[Figure]

**Figure 18.** Scorecard-type comparison for LIS/Noah-MPv4.0.1 and LIS/Noah-MPv5.0 model performance in simulating key surface variables evaluated against the reference datasets used in this study based on the ILAMB tool.

2. I would also suggest that more details be provided on the use of the Github submodule mechanism to streamline synchronization, or at least a reference on how this works and helps keeping versions in sync.

Response: Thank you for the suggestion. The detailed description of the GitHub submodule process is here: https://gist.github.com/gitaarik/8735255. We have also included additional brief explanations in Section 2.2 as follows:

     *"The GitHub submodule mechanism (https://gist.github.com/gitaarik/8735255) allows (1) separated source code maintenance and updates for Noah-MP (by the Noah-MP team) and LIS (by the NASA/LIS team), and (2) convenient updates of Noah-MP inside LIS by updating the submodule link to a newer Noah-MP GitHub tag/branch version."*

3. In many figures, grids with statistically significant differences are shown with gray dots. However, the technique used to assess whether the differences are statistically significant is not explained in the text. Please provide more details on the method used.

Response: Thank you for the suggestion. We applied the t-test to daily time series over each grid to compute statistical significance based on the widely-used SciPy python package (https://docs.scipy.org/doc/scipy/reference/generated/scipy.stats.ttest_ind.html). We have included this explanation in all figure captions where necessary as follows:

"*Grids with statistically significant differences (p < 0.05) are shown with gray dots in panels (d)-(f). The statistical significance over each grid is computed using daily time series and the t-test method.*"

4. Finally, I have read the comments made by Anonymous Referee #1 and agree that the discrepancy between model differences for soil moisture and LH needs to be investigated further. For the CONUS domain, I suggest looking at each season separately, as well as separating the evaporation and transpiration components of evapotranspiration if this is possible in LIS. Even if a comparison to observations of the two components is not possible, it could provide useful information as to the origin of the differences.

Response: Thank you for the suggestion. We have added analyses of modeled ET component (soil evaporation, plant transpiration, and canopy evaporation of intercepted water) for each season and their comparison with GLEAM data. For other discussions related to the soil moisture and LH issue, please see our responses to the comments by Referee #1 for details. In particular, we would like to highlight that the ET observational data products have large uncertainty which would confound the model evaluation. For example, our additional model evaluation against FLUXCOM-X-BASE data product shows opposite signs of model LH biases over many regions compared to the results evaluated against GLEAM. Here, to respond to the ET component evaluation over the CONUS domain raised by this specific comment, we have added the following figures and discussions in the Discussion section (Section 5) as follows:

"*The modeled LH and soil moisture assessments in Section 4 indicate a slightly higher soil moisture but lower LH over some mid-latitude (e.g., the eastern U.S.) and the tropics in LIS/Noah-MPv5.0 compared to LIS/Noah-MPv4.0.1. To investigate this seemingly contradictory signals, we conducted a series of tests and analysis. ... In addition, we quantified the differences in each of the modeled ET components between the two model versions and their biases by comparing with the GLEAM data. Using the CONUS region as an example, we find that the lower LH in LIS/Noah-MPv5.0 over the eastern U.S. is mainly caused by the lower plant transpiration and soil evaporation compared to LIS/Noah-MPv4.0.1, which exceed the higher canopy-intercepted water evaporation (Figures S13-S15). The slightly lower LH in LIS/Noah-MPv5.0 over the western U.S. is dominated by the lower plant transpiration and canopy-intercepted water evaporation, which outweigh the higher soil evaporation. These patterns are generally consistent throughout the seasons (Figures S16-18), with stronger signals for plant transpiration and soil evaporation in spring and summer due to warmer temperature and higher solar radiation. Thus, the slightly higher soil moisture appears to be a result of the lower total ET in LIS/Noah-MPv5.0 compared to LIS/Noah-MPv4.0.1.*"

[Figure]

**Figure S13**. Comparison of latent heat flux (W/m²) due to soil evaporation between the GLEAM data and LIS/Noah-MP simulations driven by the NLDAS-2 forcing over the CONUS averaged during 2018-2021: (a) GLEAM3.8a data, (b) LIS/Noah-MPv4.0.1 simulation, (c) LIS/Noah-MPv5.0 simulation, (d) LIS/Noah-MPv4.0.1 biases (model minus GLEAM), (e) LIS/Noah-MPv5.0 biases (model minus GLEAM), and (f) differences between LIS/Noah-MPv5.0 and LIS/Noah-MPv4.0.1 simulations. Grids with statistically significant differences ($p < 0.05$) are shown with gray dots in panels (d)-(f). The statistical significance over each grid is computed using daily time series and the T-test method. The global mean value is also provided in the lower right of each panel. See Figure S16 for seasonal plots.

[Figure]

**Figure S14**. Same as Figure S13 but for plant transpiration. See Figure S17 for seasonal plots.

[Figure]

**Figure S15**. Same as Figure S13 but for canopy-intercepted water evaporation. See Figure S18 for seasonal plots.

[Figure]

**Figure S16**. Same as Figure S13 but for seasonal results: (a-d) DJF, (e-h) MAM, (i-l) JJA, and (m-p) SON.

[Figure]

**Figure S17**. Same as Figure S14 but for seasonal results: (a-d) DJF, (e-h) MAM, (i-l) JJA, and (m-p) SON.

[Figure]

**Figure S18**. Same as Figure S15 but for seasonal results: (a-d) DJF, (e-h) MAM, (i-l) JJA, and (m-p) SON.

---

## Referee Report (RR1)

The authors have thoroughly addressed the comments and concerns that were raised during the previous round of reviews through the inclusion of several new analyses as well as datasets. By doing so, they have resolved the previous weaknesses of their manuscript and I think that the manuscript is now in good shape to be accepted for publication.